# EquiPocket: an E(3)-Equivariant Geometric Graph Neural Network for Ligand Binding Site Prediction

## Abstract

Predicting the binding sites of target proteins plays a fundamental role in drug discovery. Most existing deep-learning methods consider a protein as a 3D image by spatially clustering its atoms into voxels and then feed the voxelized protein into a 3D CNN for prediction. However, the CNN-based methods encounter several critical issues: 1) defective in representing irregular protein structures; 2) sensitive to rotations; 3) insufficient to characterize the protein surface; 4) unaware of protein size shift. To address the above issues, this work proposes EquiPocket, an E(3)-equivariant Graph Neural Network (GNN) for binding site prediction. In particular, EquiPocket consists of the three modules: the first one to extract local geometric information for each surface atom, the second one to model both the chemical and spatial structure of protein and the last one to capture the geometry of the surface via equivariant message passing over the surface atoms. We further propose a dense attention output layer to alleviate the effect incurred by the variable protein size. Extensive experiments on several representative benchmarks demonstrate the superiority of our framework to the state-of-the-art methods. Related codes can be found at the anonymous link [1].

## 1 Introduction

Nearly all biological and pharmacological processes in living systems involve interactions between receptors (*i.e.* target proteins) and ligands (*i.e.* small molecules or other proteins) [40]. These interactions take place at specific regions that are referred to as binding sites/pockets on the target protein structures. Predicting the ligand binding sites via in-silico algorithms forms an indispensable and even the first step for various tasks, including docking [57; 58; 34] and drug molecule design [53].

Through the past years, various computational methods for binding site detection have emerged, broadly categorized [35] into geometry-based[32; 16; 55; 30; 6; 11], probe-based [28; 29; 38; 12], and template-based methods [5; 50]. These methods exploit hand-crafted algorithms guided by domain knowledge or external templates, leading to insufficient expressivity in representing proteins. Motivated by the breakthrough of deep learning in a variety of fields, Convolutional Neural Networks (CNNs) have been applied successfully for the binding site prediction [21]. Typical works include DeepSite [20], DeepPocket [3], DeepSurf [37], etc. The CNN-based methods consider a protein as a 3D image by spatially clustering its atoms into the nearest voxels, and then model the binding site prediction as a object detection problem or a semantic segmentation task on 3D grids. These CNN-based methods have demonstrated superiority over traditional learning-based approaches and tend to achieve top performance on various public benchmarks [37].

In spite of the impressive progress, existing CNN-based methods still encounter several issues:

**Issue 1.** Defective in leveraging regular voxels to model the proteins of irregular shape. First, a considerable number of voxels probably contain no atom due to the uneven spatial distribution of protein atoms, which yields unnecessary redundancy in computation and memory. Moreover, the voxelization is usually constrained within a fixed-size (*e.g.* 70Å × 70Å × 70Å ) [20; 47]. The outside atoms will be directly discarded, resulting in incomplete and inaccurate modeling for large proteins.

**Issue 2.** Sensitive to rotations. The CNN-based methods rely on fixed coordinate bases for discretizing proteins into 3D grids. When rotating the protein, the voxelization results could be dis-

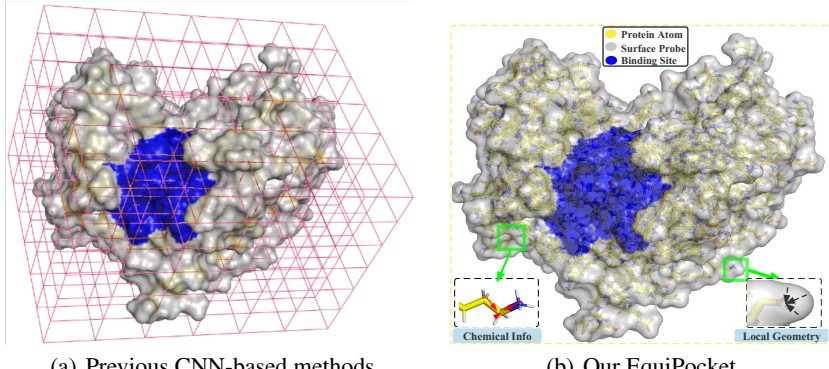

(a) Previous CNN-based methods      (b) Our EquiPocket

Figure 1: Illustrative comparison between previous CNN-based methods and our EquiPocket.

tinct, affecting predicted binding sites. This contradicts the fact that any protein rotation keeps the binding sites invariant. While it can be alleviated by local grid [37] or augmenting data with random rotations [39; 47], which yet is data-dependent and unable to guarantee rotation invariance in theory.

**Issue 3.** Insufficient to characterize the geometry of the protein surface. Surface atoms comprise the major part of the binding pocket, which should be elaborately modeled. In the CNN-based methods, surface atoms are situated within voxels surrounded by empty voxels, which somehow encodes the surface geometry. Nevertheless, such information is too coarse to depict how surface atoms interact and what their local geometry is. Indeed, the description of surface atoms is purely driven by the geometric shape of the solvent-accessible surface of the protein [41] (Figure 1(b)), which, unfortunately, is less explored in current learning-based works.

**Issue 4.** Unaware of protein size shift. In practical scenarios, the size of the proteins varies greatly across different datasets. It requires the deep learning model we apply to be well generalizable and adaptive, so that it is able to overcome the distribution shift incurred by the variable protein size. However, this point is not seriously discussed previously.

To address the above issues, this paper proposes to apply Graph Neural Networks (GNNs) [22; 7; 44] instead of CNNs to represent proteins. By considering atoms as nodes, interactions as edges, GNNs are able to encode the irregular protein structures. More importantly, a recent line of researches [44; 18; 15] has enhanced GNNs by encapsulating E(3) equivariance/invariance with respect to translations/rotations; in this way, equivariant GNNs yield outputs that are independent of the choice of the coordinate systems. That being said, trivially applying equivariant GNNs for the binding site prediction task is still incapable of providing desirable performance, and even achieves worse accuracy than the CNN-based counterparts. By looking into their design, equivariant GNNs naturally cope with the first two issues as mentioned above, yet leave the other two unsolved. To this end, we make the contributions as follows:

1) To the best of our knowledge, we are the first to apply an E(3)-equivariant GNN for binding site prediction, which is dubbed **EquiPocket**. In contrast to conventional CNN-based methods, EquiPocket is free of the voxelization process, able to model irregular protein structures by nature, and insensitive to any Euclidean transformation, thereby addressing Issue 1 and 2.

2) EquiPocket consists of three modules: the first one to extract local geometric information for each surface atom with the help of the solvent-accessible surface technique [41], the second one to model both the chemical and the spatial structures of the protein, and the last one to capture the comprehensive geometry of the surface via equivariant message passing over the surface atoms. The first and the last modules are proposed to tackle Issue 3.

3) To alleviate the effect by protein size shift in Issue 4, we further propose a novel output layer called *dense attention output layer*, which enables us to adaptively balance the scope of the receptive field for each atom based on the density distribution of the neighbor atoms.

4) Extensive experiments demonstrate the superiority of our framework to the state-of-the-art methods in prediction accuracy. The design of our model is sufficiently ablated as well.

It is worth to mention that some researchers have adopted typical GNNs for protein pocket detection and other relevant tasks [51; 36; 13]. However, all these methods are non-equivariant and not geometry-aware, and can solely extract the structure information of the target protein, leading to worse performance than 3D CNN-based methods, which will be shown in our experiments.

## 2 NOTATIONS AND DEFINITIONS

**Protein Graph.** A protein such as the example in Figure 1(b) is denoted as a graph $\mathcal{G}_P = (\mathcal{V}_P, \mathcal{E}_C, \mathcal{E}_D)$, where $\mathcal{V}_P = \{v_0, ..., v_N\}$ forms the set of $N$ atoms, $\mathcal{E}_C$ represents the chemical-bond edges, and $\mathcal{E}_D$ collects the spatial edges between any two atoms if their spatial distance is less than a cutoff $\theta > 0$. In particular, each node (*i.e.* atom) is associated with a feature $(\boldsymbol{x}_i, \boldsymbol{c}_i)$, where $\boldsymbol{x}_i \in \mathbb{R}^3$ denotes the 3D coordinates and $\boldsymbol{c}_i \in \mathbb{R}^5$ is the chemical feature.

**Surface Probe Set.** The surface geometry of a protein is of crucial interest for binding site detection. By employing the open source MSMS [43], as showed in Figure 2, we move a probe (the grey circle) of a certain radius along the protein to calculate the Solvent Accessibility Surface (SAS) and Solvent Excluded Surface (SES) [31]. The resulting coordinates of probe are considered as surface probes. Here we define the set of surface probes, by $\mathbb{S} = \{s_0, ..., s_M\}$, $M \gg N$. Each surface probe $s_i$ corresponds to $(\boldsymbol{x}_i, p_i)$, where $\boldsymbol{x}_i \in \mathbb{R}^3$ represents the 3D coordinates of $s_i$ and $p_i \in \mathcal{V}_P$ indicates the index of the nearest protein atom in $\mathcal{V}_P$ to $s_i$.

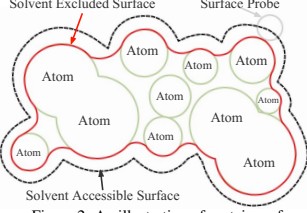

Figure 2: An illustration of protein surface.

**Protein Surface Graph.** Referring to the surface probes defined above, we collect all the nearest protein atoms $p_i$ of the surface probes, forming the surface graph $\mathcal{G}_S = (\mathcal{V}_S, \mathcal{E}_S)$, and clearly $\mathcal{G}_S \subseteq \mathcal{G}_P$. Notably, the edges of the surface graph, *i.e.* , $\mathcal{E}_S$ is only composed of spatial edges from $\mathcal{E}_D$, since those chemical edges are mostly broken among the extracted atoms.

**Equivariance and Invariance.** In 3D space, the symmetry of the physical laws requires the detection model to be equivariant with respect to arbitrary coordinate systems [15]. In form, suppose $\boldsymbol{X}$ to be 3D geometric vectors (positions, velocities, etc) that are steerable by E(3) group (rotations/translations/reflections), and $\boldsymbol{h}$ non-steerable features. The function $f$ is E(3)-equivariant, if for any transformation $g \in \mathrm{E}(3)$, $f(g \cdot \boldsymbol{X}, \boldsymbol{h}) = g \cdot f(\boldsymbol{X}, \boldsymbol{h}), \forall \boldsymbol{X} \in \mathbb{R}^{3 \times m}, \boldsymbol{h} \in \mathbb{R}^d$. Similarly, $f$ is invariant if $f(g \cdot \boldsymbol{X}, \boldsymbol{h}) = f(\boldsymbol{X}, \boldsymbol{h})$. The group action $\cdot$ is instantiated as $g \cdot \boldsymbol{X} := \boldsymbol{X} + \boldsymbol{b}$ for translation $\boldsymbol{b} \in \mathbb{R}^3$ and $g \cdot \boldsymbol{X}(t) := \boldsymbol{O}\boldsymbol{X}$ for rotation/reflection $\boldsymbol{O} \in \mathbb{R}^{3 \times 3}$.

**Problem Statement.** Given a protein $\mathcal{G}_P$, its surface probes $\mathbb{S}$, and constructed surface graph $\mathcal{G}_S$, our goal is to learn an E(3)-invariant model $f(\mathcal{G}_P, \mathbb{S}, \mathcal{G}_S)$ to predict the atoms $\mathcal{V}_B$ of the binding site.

## 3 THE PROPOSED METHODOLOGY

Figure 4 illustrates the overall framework of our EquiPocket, which consists of three modules: the *local geometric modeling module* § 3.1 that focuses on extracting the geometric information of each surface atom, the *global structure modeling module* § 3.2 to characterize both the chemical and spatial structures of the protein, and the *surface message passing module* § 3.3 which concentrates on capturing the entire surface geometry based on the extracted information by the two former modules. The training losses are also presented. We defer the pseudo codes of EquiPocket to Appendix 1.

### 3.1 LOCAL GEOMETRIC MODELING MODULE

This subsection presents how to extract the local geometric information of the protein surface $\mathcal{G}_S$, with the help of surface probes $\mathbb{S}$. The local geometry of each protein atom closely determines if the region nearby is appropriate or not to become part of binding sites. We adopt the surrounding surface probes of each protein surface atom to describe the local geometry. To be specific, for every surface atom $i \in \mathcal{V}_S$, its surrounding surface probes are returned by a subset of $\mathbb{S}$, namely, $\mathbb{S}_i = \{s_j = (\boldsymbol{x}_j, p_j) \in \mathbb{S} \mid p_j = i\}$, where $p_j$, indicates the nearest protein atom. We now construct the geometric information based on $\mathbb{S}_i$. We denote the center/mean of all 3D coordinates in $\mathbb{S}_i$ as $\bar{\boldsymbol{x}}_i$. For each surrounding surface probe $s_j \in \mathbb{S}_i$, we first search its two nearest surface probes from $\mathbb{S}$ as $s_{j_1}$ and $s_{j_2}$, and then calculate the following relative position vectors:

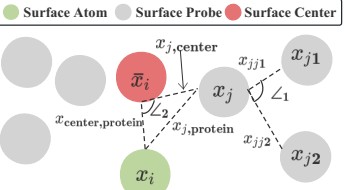

Figure 3: An illustration of local geometric features.

$$\boldsymbol{x}_{jj_1} = \boldsymbol{x}_j - \boldsymbol{x}_{j_1}, \boldsymbol{x}_{jj_2} = \boldsymbol{x}_j - \boldsymbol{x}_{j_2},$$
$$\boldsymbol{x}_{j,\text{center}} = \boldsymbol{x}_j - \bar{\boldsymbol{x}}_i, \boldsymbol{x}_{j,\text{protein}} = \boldsymbol{x}_j - \boldsymbol{x}_i, \boldsymbol{x}_{\text{center,protein}} = \bar{\boldsymbol{x}}_i - \boldsymbol{x}_i. \tag{1}$$

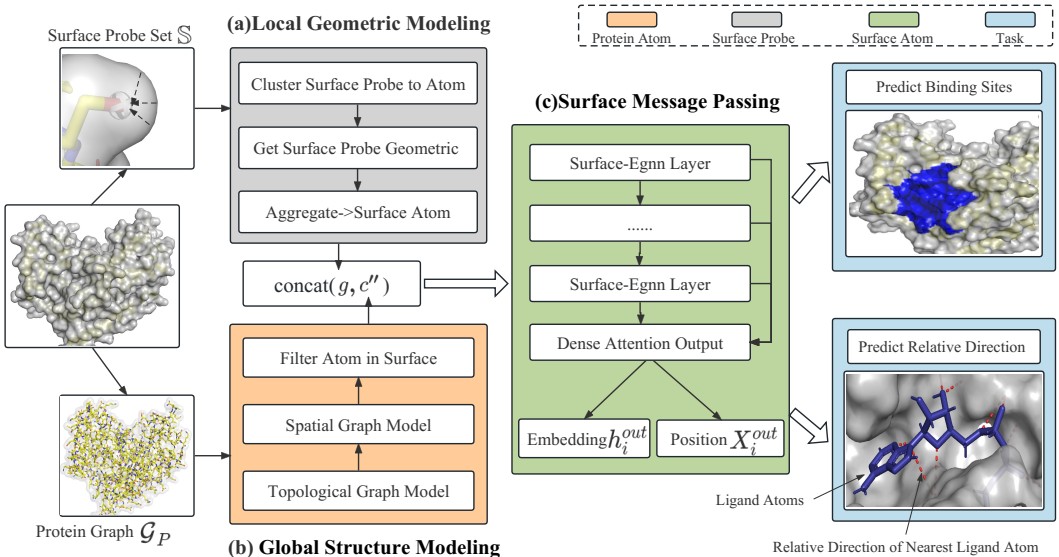

Figure 4: An illustration of the scheme of our EquiPocket framework.

We further derive the following scalars upon Eq. 1:

$$\boldsymbol{g}(s_j) := [\|\boldsymbol{x}_{jj_1}\|_2, \|\boldsymbol{x}_{jj_2}\|_2, \angle_1, \|\boldsymbol{x}_{j,\text{center}}\|_2, \|\boldsymbol{x}_{j,\text{protein}}\|_2, \|\boldsymbol{x}_{\text{protein,center}}\|_2, \angle_2], \tag{2}$$

where the angels are computed by $\angle_1 = \frac{\boldsymbol{x}_{jj_1} \cdot \boldsymbol{x}_{jj_2}}{\|\boldsymbol{x}_{jj_1}\|_2 \|\boldsymbol{x}_{jj_2}\|_2}$ and $\angle_2 = \frac{\boldsymbol{x}_{j,\text{center}} \cdot \boldsymbol{x}_{\text{center,protein}}}{\|\boldsymbol{x}_{j,\text{center}}\|_2 \|\boldsymbol{x}_{\text{center,protein}}\|_2}$; here the operator $\cdot$ defines the inner-product between two vectors. Basically, as displayed in Figure 3, the first three quantities in $\boldsymbol{g}(s_j)$ depict how the nearby surface probes are arranged around $s_j$, and the last four ones describe where $s_j$ is located within the global region of $\mathbb{S}_i$.

We aggregate the geometric information $\boldsymbol{g}(s_j)$ over all surface probes in $\mathbb{S}_i$ and obtain a readout descriptor for surface atom $i$ as

$$\boldsymbol{g}_i = [\text{Pooling}(\{\text{MLP}(\boldsymbol{g}(s_j))\}_{s_j \in \mathbb{S}_i}), \text{MLP}(\text{Pooling}\{(\boldsymbol{g}(s_j))\}_{s_j \in \mathbb{S}_i})] \tag{3}$$

Here, MLP denotes multi-layer perceptron, and the function Pooling is implemented as a concatenation of mean pooling and max pooling throughout our experiments. The front part in Eq. 3 is used to gather local geometric features, while the latter part attempts to compute the global size of surrounding surface probes. Notably, the geometric descriptor $\boldsymbol{g}_i$ is E(3)-invariant.

### 3.2 GLOBAL STRUCTURE MODELING MODULE

This module aims at processing the information of the whole protein $\mathcal{G}_P$, including atom type, chemical bonds, relevant spatial positions, etc. Although the binding pocket is majorly comprised of surface atoms, the global structure of the protein in general influences how the ligand is interacted with and how the pocket is formulated, which should be modeled. We fulfil this purpose via two concatenated processes: chemical-graph modeling and spatial-graph modeling.

The chemical-graph modeling process copes with the chemical features $\{\boldsymbol{c}_i\}_{i \in \mathcal{V}_P}$ and the chemical interactions $\mathcal{E}_C$ of the protein graph. For each atom in the protein, its chemical type, the numbers of electrons around, and the chemical bonds connected to other atoms are important clues to identify the interaction between the protein and the ligand [57]. We employ typical GNNs [23; 52; 42] to distill this type of information. Formally, we proceed:

$$\{\boldsymbol{c}'_i\}_{i \in \mathcal{V}_P} = \text{GNN}(\{\boldsymbol{c}_i\}_{i \in \mathcal{V}_P}, \mathcal{E}_C), \tag{4}$$

where $\boldsymbol{c}'_i$ is the updated chemical feature for atom $v_i$. While various GNNs can be used in Eq. 4, here we implement GAT [52] given its desirable performance observed in our experiments.

The spatial-graph modeling process further involves the 3D coordinates $\{\boldsymbol{x}_i\}_{i \in \mathcal{V}_P}$ to better depict the spatial interactions $\mathcal{E}_D$ within the protein. Different from chemical features $\boldsymbol{c}'_i$, the 3D coordinates provide the spatial position of each atom and reflect the pair-wise distances in 3D space, which is helpful for physical interaction modeling. We leverage EGNN [44] as it conforms to E(3) equivariance/invariance and achieves promising performance on modeling spatial graphs. Specifically,

we process EGNN as follows:

$$\{c_i''\}_{i \in \mathcal{V}_P} = \text{EGNN}(\{x_i, c_i'\}_{i \in \mathcal{V}_P}, \mathcal{E}_D). \tag{5}$$

Here, we only reserve the invariant output (*i.e.* , $c_i''$) and have discarded the equivariant output (*e.g.* updated 3D coordinates) of EGNN, since the goal of this module is to provide invariant features. We select the updated features of the surface atoms $\mathcal{V}_S$, which will be fed into the module in § 3.3.

## 3.3 SURFACE MESSAGE PASSING MODULE.

Given the local geometric features $\{g_i\}_{i \in \mathcal{V}_S}$ from § 3.1, and the globally-encoded features of the surface atoms $\{c_i''\}_{i \in \mathcal{V}_S}$ from § 3.2, the module in this subsection carries out equivariant message passing on the surface graph $\mathcal{G}_S$ to renew the entire features of the protein surface. We mainly focus on the surface atoms here, because firstly the surface atoms are more relevant to the binding sites than the interior atoms, and secondly the features $\{c_i''\}_{i \in \mathcal{V}_S}$ that are considered as the input have somehow encoded the information of the interior structure via the processes in 3.2.

**Surface-EGNN.** During the $l$-th layer message passing, each node is associated with an invariant feature $h_i^{(l)} \in \mathbb{R}^{m_l}$ and an equivariant double-channel matrix $X_i^{(l)} \in \mathbb{R}^{3 \times 2}$. We first concatenate $c_i''$ with $g_i$ as the initial invariant feature, $h_i^{(0)} = [c_i'', g_i]$. The equivariant matrix $X_i^{(0)}$ is initialized by the 3D coordinates of the atom and the center of its surrounding surface probes, that is, $X_i^{(0)} = [x_i, \bar{x}_i]$. We update $h_i^{(l)} \in \mathbb{R}^{d_l}$ and $X_i^{(l)} \in \mathbb{R}^{3 \times 2}$ synchronously to unveil both the topological and geometrical patterns. Inspired from EGNN [44] and its multi-channel version GMN [18], we formulate the $l$-th layer for each surface atom $i \in \mathcal{V}_S$ as:

$$m_{ij} = \phi_m\left(h_i^{(l)}, h_j^{(l)}, f_x(X_i^{(l)}, X_j^{(l)}), e_{ij}\right), h_i^{(l+1)} = \phi_h\left(h_i^{(l)}, \sum_{j \in \mathbb{N}(i)} m_{ij}\right) \tag{6}$$

$$X_i^{(l+1)} = X_i^{(l)} + \frac{1}{|\mathbb{N}(i)|} \sum_{j \in \mathcal{N}(i)} (x_{i,1}^{(l)} - x_{j,1}^{(l)}) \phi_x(m_{ij}), \tag{7}$$

where the functions $\phi_m, \phi_h, \phi_x$ are MLPs, $x_{i,1}$ ($x_{j,1}$) denotes the first channel of $X_i$ ($X_j$), $\mathbb{N}(i)$ denotes the spatial neighbors of node $i$, $|\cdot|$ counts the size of the input set, and the invariant message $m_{ij}$ from node $j$ to $i$ is employed to update the invariant feature $h_i^{(l+1)}$ via $\phi_h$ and the equivariant matrix $X_i^{(l+1)}$ via the aggregation of the relative position $x_{i,1}^{(l)} - x_{j,1}^{(l)}$ multiplied with $\phi_x$.

As a core operator in the message passing above, the function $f_x(X_i, X_j)$ is defined as follows:

$$f_x(X_i, X_j) := [\|x_{ij}\|_2, \|x_{ci}\|_2, \|x_{cj}\|_2, \angle_{ci,ij}, \angle_{cj,ij}, \angle_{ci,cj}], \tag{8}$$

where, the relative positions are given by $x_{ij} = x_{i,1} - x_{j,1}$, $x_{ci} = x_{i,2} - x_{i,1}$ and $x_{cj} = x_{j,2} - x_{j,2}$; the angles $\angle_{ci,ij}, \angle_{cj,ij}, \angle_{ci,cj}$ are defined as the inner-products of the corresponding vectors denoted in the subscripts, *e.g.* , $\angle_{ci,ij} = \frac{x_{ci} \cdot x_{ij}}{\|x_{ci}\|_2 \|x_{ij}\|_2}$. Through the design in Eq. 8, $f_x(X_i, X_j)$ elaborates the critical information (including relative distances and angles) around the four points: $x_{i,1}, x_{i,2}, x_{j,1}, x_{j,2}$, which largely characterizes the geometrical interaction between the two input matrices. Nicely, $f_x(X_i, X_j)$ is invariant, ensuring the equivariance of Surface-EGNN.

**Dense Attention Output Layer.** Conventionally, we can apply the output of the final layer, *i.e.* , $(h_i^{(L)}, X_i^{(L)})$ to estimate the binding site. Nevertheless, such flat output overlooks the discrepancy of size and shape between different proteins. As showed in Figure 6(b), for small or densely-connected proteins, the receptive field of each node will easily cover most nodes after a small number of message-passing layers, and excessive message passing will lead to over-smoothing [17] that will incurs performance detriment. For large or sparsely-connected proteins, on the contrary, insufficient message passing can hardly attain the receptive field with a desirable scope, which will also decrease the performance. It thus requires us to develop an adaptive mechanism to balance the message passing scope between different proteins. We propose the *dense attention output layer* (showed in Figure 5) to achieve this goal.

Intuitively, for each target atom, the spatial distribution of neighbors can reflect the density of spatial connections around. This motivates us to calculate the proportion of the atoms with different distance ranges. we take $\theta$ as the distance unit to create the spatial graph and compute by:

$$n_i^{(l)} = \frac{|\{j \in \mathcal{V}_P \mid 0 \leq \|x_i - x_j\|_2 < l\theta\}|}{N_P}, \tag{9}$$

where, the proportion is evaluated within the distance range $[0, l\theta]$, $N_P = |\mathcal{V}_P|$, and the neighbor hop $l \in \mathbb{Z}^+$. We collect the proportions of all hops from 0 to $L$, yielding the proportion vector $\boldsymbol{n}_i = [n_i^{(0)}, n_i^{(1)}, \cdots, n_i^{(L)}, N_P] \in \mathbb{R}^{L+2}$ with $N_P$ plus to emphasize the total number of the protein atoms. Clearly, $\boldsymbol{n}_i$ contains rich information of the spatial density, and we apply it to determine the importance of different layers, by producing the attention $\boldsymbol{a}_i = \text{Sigmoid}(\phi_a(\boldsymbol{n}_i))$. Here, $\phi_a$ is an MLP with the number of output channels as $L + 1$, the Sigmoid function[1] is applied for each channel, implying that $\boldsymbol{a}_i \in (0, 1)^{L+1}$.

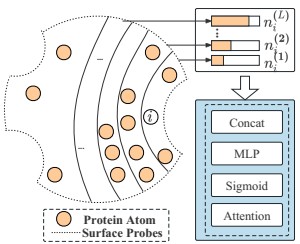

Figure 5: An illustration of Dense Attention.

Subsequently, we multiply the hidden feature of corresponding layer with each channel of attention vector. The results are then concatenated into a vector denoted as $\boldsymbol{h}_i^{\text{out}}$. To retain translation equivariance, we calculate the mean coordinates of all layers as $\boldsymbol{X}_i^{\text{out}}$:

$$\boldsymbol{h}_i^{\text{out}} = \text{Concat}(a_{i0}\boldsymbol{h}_i^{(0)}, ..., a_{iL}\boldsymbol{h}_i^{(L)}), \quad \boldsymbol{X}_i^{\text{out}} = \frac{1}{L+1}\sum_{l=0}^{L}\boldsymbol{X}_i^{(l)}, \tag{10}$$

where $a_{il}$ is the $l$-th channel of $\boldsymbol{a}_i$. By making use of Eq. 10, the learnable attentions enable the model to adaptively balance the importance of different layers for different input proteins. We will illustrate the benefit of the proposed strategy in our experiments.

### 3.4 OPTIMIZATION OBJECTIVE

**Binding Sites Prediction.** We set $y_i = 1$ if a surface atom $i$ is within 4Å to any ligand atom [37] and compute $\hat{y}_i = \textbf{Sigmoid}(\text{MLP}(\boldsymbol{h}_i^{\text{out}}))$ as the probability of being a part of binding site according its dense embedding $h_i^{\text{out}}$. The loss $\mathcal{L}_b$ for this task is computed with Dice loss [20; 21].

**Relative Direction Prediction.** Beyond the CNN-based methods, our EquiPocket is an E(3)-equivariant model, which can not only output the embedding $\boldsymbol{h}_i^{\text{out}}$ but also the coordinate matrix $\boldsymbol{X}_i^{\text{out}}$ (with initial position vector $\boldsymbol{x}_i$). To enhance our framework for gathering local geometric features, we further leverage the position vector $\boldsymbol{m}_i$ to compute the relative direction of its nearest ligand atom by $\boldsymbol{d}_i = \frac{\boldsymbol{m}_i - \boldsymbol{x}_i}{\|\boldsymbol{m}_i - \boldsymbol{x}_i\|_2}$, which is predicted as $\hat{\boldsymbol{d}}_i = \frac{\boldsymbol{x}_i^{\text{out}} - \boldsymbol{x}_i}{\|\boldsymbol{x}_i^{\text{out}} - \boldsymbol{x}_i\|_2}$. The task loss $\mathcal{L}_d$ is computed with cosine loss. We compute the eventual loss by $\mathcal{L} = \mathcal{L}_b + \mathcal{L}_d$.

## 4 EXPERIMENTS

In this section, we will perform experiments on various datasets to assess the performance of our framework and individual modules. Related codes and resources for our experiments can be found at the anonymous link [1].

### 4.1 EXPERIMENTAL SETTINGS

**Dataset.** scPDB [10] is the famous dataset for binding site prediction, which contains the protein, ligand, and 3D cavity structure generated by VolSite [9]. The 2017 release is used for training and cross-validation. PDBbind [54] is a commonly used dataset for researching protein-ligand complex, which contains the 3D structures of proteins, ligands, binding sites, and binding affinity results determined in the laboratory. We use the v2020 release for evaluation. COACH 420 and HOLO4K [27] are two test datasets for binding site prediction. We use the mlig subsets for evaluation [27; 37; 3]. The data summary and preparation process are detailed in Appendix A.4.1 and A.5.

**Target of Binding Sites.** Following [37], the protein atoms within 4Å of any ligand atom are set as positive and negative otherwise. After obtaining the probability that an atom is a candidate binding site, we use the mean-shift algorithm [8] to predict the binding site center, which can determine the number of clusters on its own (details in Appendix A.5.2). The CNN-based methods [20; 3; 47] mark a grid as positive if its distance from the binding site's geometric center is less than 4Å.

**Evaluation Metrics.** We take the metrics including **DCC** (Distance between the predicted binding site center and the true binding site center), **DCA** (Shortest distance between the predicted binding

---

[1]Note that the sum of all channels of $\boldsymbol{a}_i$ is unnecessarily equal to 1, since the Sigmoid function instead of the previously-used SoftMax function is applied here.

Table 1: Experimental and ablation results of baseline models and our framework.[a]

| Methods | Type | Param (M) | Failure Rate ↓ | COACH420 DCC↑ | DCA↑ | HOLO4K DCC↑ | DCA↑ | PDBbind2020 DCC↑ | DCA↑ |
|---|---|---|---|---|---|---|---|---|---|
| Fpocket[b] | Geometric-based | \ | **0.000** | 0.228 | 0.444 | 0.192 | 0.457 | 0.253 | 0.371 |
| DeepSite[b] | 3D-CNN | 1.00 | \ | \ | 0.564 | \ | 0.456 | \ | \ |
| Kalasanty[b] | | 70.64 | 0.120 | 0.335 | 0.636 | 0.244 | 0.515 | 0.416 | 0.625 |
| DeepSurf[b] | | 33.06 | 0.054 | 0.386 | **0.658** | 0.289 | 0.635 | 0.510 | 0.708 |
| GAT | Topological Graph | **0.03** | 0.11 | 0.039(0.005) | 0.130(0.009) | 0.036(0.003) | 0.110(0.010) | 0.032(0.001) | 0.088(0.011) |
| GCN | | 0.06 | 0.163 | 0.049(0.001) | 0.139(0.010) | 0.044(0.003) | 0.174(0.003) | 0.018(0.001) | 0.070(0.002) |
| GAT + GCN | | 0.08 | 0.31 | 0.036(0.009) | 0.131(0.021) | 0.042(0.003) | 0.152(0.020) | 0.022(0.008) | 0.074(0.007) |
| GCN2 | | 0.11 | 0.466 | 0.042(0.098) | 0.131(0.017) | 0.051(0.004) | 0.163(0.008) | 0.023(0.007) | 0.089(0.013) |
| SchNet | Spatial Graph | 0.49 | 0.14 | 0.168(0.019) | 0.444(0.020) | 0.192(0.005) | 0.501(0.004) | 0.263(0.003) | 0.457(0.004) |
| Egnn | | 0.41 | 0.270 | 0.156(0.017) | 0.361(0.020) | 0.127(0.005) | 0.406(0.004) | 0.143(0.007) | 0.302(0.006) |
| EquiPocket-L | Ours | 0.15 | 0.552 | 0.070(0.009) | 0.171(0.008) | 0.044(0.004) | 0.138(0.006) | 0.051(0.003) | 0.132(0.009) |
| EquiPocket-G | | 0.42 | 0.292 | 0.159(0.016) | 0.373(0.021) | 0.129(0.005) | 0.411(0.005) | 0.145(0.007) | 0.311(0.007) |
| EquiPocket-LG | | 0.50 | 0.220 | 0.212(0.016) | 0.443(0.011) | 0.183(0.004) | 0.502(0.008) | 0.274(0.004) | 0.462(0.005) |
| EquiPocket | | 1.70 | 0.051 | **0.423(0.014)** | 0.656(0.007) | **0.337(0.006)** | **0.662(0.007)** | **0.545(0.010)** | **0.721(0.004)** |

[a] The standard deviation of each index is indicated in brackets. The result of 5-fold for EquiPocket is shown in Appendix A.10.1.
[b] We use their published pretrained models or published results, details in Appendix A.5.5.

site center and any grid of the ligand) and **Failures Rate** (Sample rate without any predicted binding site center). Success rate is determined for samples with the DCC(DCA) values below a predetermined threshold. Following [20; 37; 3; 47], we set the threshold to 4 Å. Details in Appendix A.5.1.

**EquiPocket Framework.** We implement our EquiPocket framework based on (GAT [52]+EGNN [44]) as our global structure modeling module. The the radius of the probe, cutoff $\theta$ and depth in our surface-egnn model are set to 1.5, 6 and 4 . To indicate the EquiPocket Framework with different modules, we adopt the following symbol as follows: i) **EquiPocket-L**: Only contain the local geometric modeling module. ii) **EquiPocket-G**: Only contain the global structure modeling module. iii) **EquiPocket-LG**: Only contain both the local geometric and global structure modeling modules. iii) **EquiPocket**: Contain all the modules.

**Baseline Models.** 1) geometric-based method(Fpocket [30]); 2) CNN-based methods (Deep-Site [20], Kalasanty [47] and DeepSurf [37]); 3) topological graph-based models (GAT [52], GCN [22] and GCN2 [7]); 4) spatial graph-based models (SchNet [46], EGNN [45]).

## 4.2 MODEL COMPARISON

In Table 1, we compared our EquiPocket framework with baseline methods mentioned above. As can be observed, the performance of the computational method Fpocket is inferior, with no failure rate, since it simply employs the geometric feature of a protein. The performance of CNN-based methods is much superior to that of the conventional method, with DCC and DCA metrics improving by more than 50 percent but requiring enormous parameter values and computing resources. However, these two early methods DeepSite and Kalasanty are hampered by protein size shift (Issue 4) and their inability to process big proteins, which may fail prediction. The recently proposed method Deepsurf employs the local-grid concept to handle any size of proteins, although CNN architecture also still results in inevitable failures.

For graph models, the poor performance of topological-graph models (GCN, GAT, GCN2) is primarily due to the fact that they only consider atom attributes and chemical bond information, ignoring the spatial structure in a protein. The performance of spatial-graph models is generally better than that of topological-graph models. EGNN model utilizes not only the properties of atoms but also their relative and absolute spatial positions, resulting in a better effect. SchNet merely updates the information of atoms based on the relative distance of atoms. We attempt to execute the Dimenet++ [24], which uses the angle info between atoms, but it requires too many computing resources, resulting in an OOM (Out Of Memory) error. However, the performance of spatial-graph model is worse than that of CNN-based and geometric-based methods because the former cannot obtain enough geometric features (Issue 3) and cannot address the protein size shift (Issue 4).

As the above results indicate, geometric info of protein surface and multi-level structure info in a protein is essential for binding site prediction. In addition, it reflects the limitations of the current GNN models, where it is difficult to collect sufficient geometric information from the protein surface or the calculation resources are too large to apply to macromolecular systems like proteins. Consequently, our EquiPocket framework is not only able to update chemical and spatial information from an atomic perspective but also able to effectively collect geometric information without excessive computing expense, resulting in a 10-20% increase in effect over previous results. Case study based on different methods is showed in Appendix A.5.7.

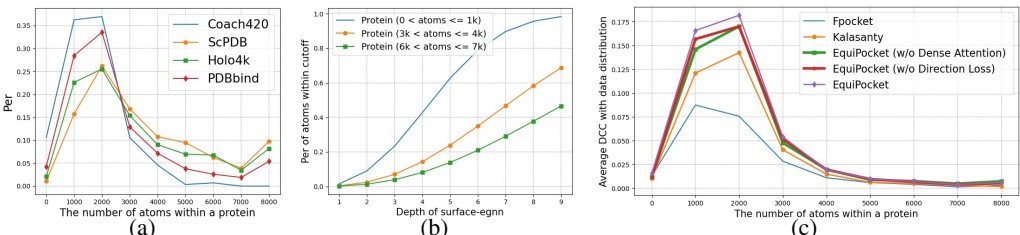

Figure 6: The protein size shift and model performances for proteins of various sizes

## 4.3 ABLATION STUDY

As shown in Table 1, we conduct ablation experiments on our EquiPocket with different modules.

**Local Geometric Modeling Module.** This module is used to extract the geometric features of protein atoms from their nearest surface probes. EquiPocket-G consists solely of this module, and the performance is negligible. There are two primary causes for this result. First, geometric information can only determine part of the binding sites. Second, it can only reflect the geometric features over a relatively small distance and cannot cover an expansive area.

**Global Structure Modeling Module.** The primary purpose of this module is to extract information about the whole protein, such as atom type, chemical bonds, relevant spatial positions, etc. We implement EquiPocket-G based on (GAT + EGNN) models, which is E(3) equivariance/invariance and has a better effect than its predecessor, EquiPocket-L. In comparison, the value of DCC increased by about 10%, and DCA increased by about 20%. This demonstrates that structure information of the whole protein is necessary for binding site prediction. In addition, when the two modules are combined as the EquiPocket-LG, the prediction effect is significantly improved, proving the complementarity of surface geometric information and global structure information.

**Surface Message Passing Module.** In the previous model, EquiPocket-LG, information was extracted solely from atoms and their closest surface probes. Nonetheless, the binding site is determined not only by the information of a single atom but also by the atoms surrounding it. Therefore, the surface message passing module is proposed to collect and update the atom's features from its neighbors. After adding this module, the performance of EquiPocket has been significantly enhanced, DCC and DCA have increased by approximately 20% on average, and the failure rate has been significantly reduced. Through the addition of multiple modules, we address the Issue 3 and the performance of our framework eventually surpasses that of the existing SOTA method, demonstrating the efficacy of our framework design.

## 4.4 PROTEIN SIZE SHIFT

As depicted in the Figure 6(a) and 6(b) that after data processing, there is a significant gap in protein size and distribution between the training dataset (scPDB) and the test dataset (COACH420, HOLO4k, PDBbind). The number of atoms within a protein ranges from hundreds to tens of thousands. As for protein distribution in datasets, scPDB has the longest average structure, followed by HOLO4k and PDBbind, with COACH420 having the shortest average protein structure. This fact will hurt model learning and generalization.

We calculate the average DCC with the distribution of various sizes proteins presented in Figure 6(c). The geometric-based method Fpocket only utilizes the geometric features of a protein surface. Therefore, its performance is superior to that of most other methods for proteins with fewer than 1,000 atoms, but its prediction effect decreases significantly as the size of the protein increases. Kalasanty is a CNN-based and learn-based method. As the number of atoms in the protein varies, the prediction effect exhibits an increasing and then a decreasing trend, which is not only influenced by the size of the protein but also has a significant correlation with the dataset's distribution. According to the train data (scPDB), the majority of proteins contain fewer than 2,000 protein atoms (as depicted in Figure 6(a)). Consequently, the model's parameters will be biased toward this protein size. In addition, for proteins with more than 8000 atoms, the prediction effect is not even as good as the geometric-based method. This is due to the fact that CNN methods typically restrict the protein space to 70Å * 70Å * 70Å, and for proteins larger than this size, the prediction frequently fails. For our EquiPocket framework, we do not need to cut the protein into grids, and we utilize both geomet-

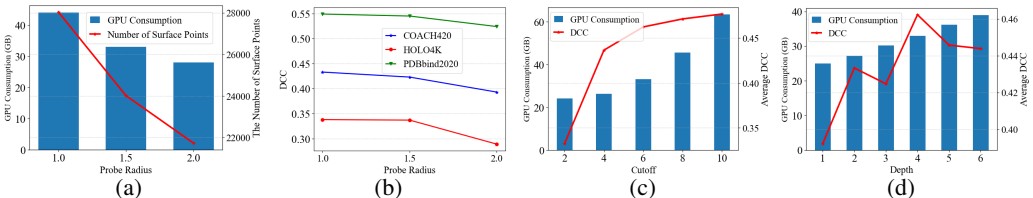

Figure 7: The influences of the probe radius of MSMS, cutoff $\theta$ and depth of surface-egnn

ric information from the surface probes and global structure information from the whole protein, so the performance for proteins of varying sizes is significantly superior to that of other methods.

**Dense Attention.** The Dense Attention is introduced in § 3.3 to reduce the negative impact caused by the protein size shift (Issue 4). As shown in 6(c), when the number of atoms contained in a protein is less than 3000, the result of the EquiPocket (w/o Dense Attention) is weaker than that of the original EquiPocket, whereas when the protein is larger, there is no significant distinction between the two models. It simply reflects the role of Dense Attention, which, by weighting the surface-egnn layer at different depths, mitigates the detrimental effect of the protein size shift.

**Direction Loss.** Direction loss is a novel task designed to improve the extraction of local geometric features. The result of the EquiPocket (w/o Direction Loss) in Figure 6(c) demonstrates conclusively that the prediction performance of small proteins with fewer than 3,000 atoms is diminished in the absence of this task, which reveals the importance of the task.

### 4.5 HYPERPARAMETERS ANALYSIS

In our EquiPocket framework, the probe radius of MSMS, the cutoff $\theta$ and depth of surface-egnn are crucial parameters that can impact performance and computational efficiency.

**Probe Radius.** We implement various radius of probe (1, 1.5, 2), which can control the number and density of surface probes. As showed in Figure 7(a) and 7(b), when reducing the radius from 1.5 to 1, the DCC accuracy shows a slight improvement. Conversely, when increasing the radius from 1.5 to 2, the DCC accuracy notably worsens. This is understandable since a smaller radius allows for a more detailed capture of geometric information around the protein surface, leading to more precise pocket detection. However, a smaller radius also results in a larger number of surface probes, increasing the GPU memory usage. To strike a balance between memory consumption and detection accuracy, we opt for the default radius value of 1.5 in our experiments. The detail results are provided in Appendix A.5.6.

**Cutoff $\theta$.** We set the depth of surface-egnn to 4 and implement various cutoff values (2, 4, 6, 8, 10). Figure 7(c) indicates that with the cutoff set to 2, the average DCC of our framework is poor, and GPU memory is relatively low (22GB). This is due to the fact that when the cutoff is small, the surface-egnn can only observe a tiny receptive field. As the cutoff increases, the performance and GPU memory continue to rise until the DCC reaches a bottleneck when the cutoff is 10, and the GPU memory reaches 62GB. Therefore, when selecting parameters for our framework, we must strike a balance between performance and efficiency.

**Depth.** The depth of surface-egnn has an immediate influences on the performance and computation cost. We set the cutoff to 6 and implement various depth (1, 2, 3, 4, 5, 6). Figure 7(d) demonstrates that as depth increases, performance steadily improves and becomes stable as GPU memory continues to expand. Because the prediction of binding sites is highly influenced by their surrounding atoms, therefore, an excessively large receptive field may not offer any benefits but will necessitate additional computing resources.

## 5 CONCLUSION

In this paper, concentrating on the ligand binding site prediction, we propose a novel E(3)-Equivariant geometric graph framework called EquiPocket, which contains the local geometric modeling module, global structure modeling module, and surface passing module to gather the surface geometric and multi-level structure features in a protein. Experiments demonstrate that our framework is highly generalizable and beneficial, and achieves superior prediction accuracy and computational efficiency compared with the existing methods.

## REPRODUCIBILITY STATEMENT

We provide the source code of our framework and the dataset information at the anonymous link [1] and supplementary materials for reproducibility.

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

## A    APPENDIX

### A.1    THE PSEUDO-CODE OF OUR EQUIPOCKET FRAMEWORK

---

**Algorithm 1:** EquiPocket

---

**Input:** Protein structure $\mathcal{G}_P$
**Output:** Candidate Binding sites and their ligandability score
 1:  Clean Structure by removing solvent, hydrogens atoms
 2:  Create the solvent accessible surface of the protein $\mathbb{S}$ use MSMS
 3:  **for** every $s_i$ in $\mathbb{S}$ **do**
 4:      Get its closed protein atom $p_i$
 5:  **end for**
 6:  Get the surface atom $\mathcal{V}_S$ according to the surface points's closed protein atom
 7:  **for** every surface atom $i \in \mathcal{V}_S$ **do**
 8:      Get their surrounding surface points set $\mathbb{S}_i$
 9:      Get the geometric embedding $\boldsymbol{g}_i$
10:  **end for**
11:  Get the global structure embedding $\boldsymbol{c}_i^{''}$ of the protein
12:  **for** every surface atom $i \in \mathcal{V}_S$ **do**
13:      Get its invariant feature $\boldsymbol{h}_i^{(0)} = [\boldsymbol{c}_i^{''}, \boldsymbol{g}_i].]$ and equivariant position matrix $\boldsymbol{X}_i^{(0)} = [\boldsymbol{x}_i, \bar{\boldsymbol{x}}_i]$
14:      Get the updated embedding $\boldsymbol{h}_i^{(l)}$ and updated coordinates $\boldsymbol{X}_i^{(l)}$ based on our surface-egnn model
15:      Get the dense embedding $\boldsymbol{h}_i$ and position $\boldsymbol{X}_i$ according to its dense attention $\boldsymbol{a}_i$
16:      predict the probability $\hat{\boldsymbol{y}}_i$ as ligandability score and the nearest ligand atom direction $\boldsymbol{d}_i$
17:  **end for**
18:  Discard protein atoms with probability less than T (T=0.5 in our experiments);
19:  Cluster the remaining protein atoms;
20:  Form binding sites and get the average ligandability score for each cluster;
21:  Rank the predicted binding sites by their ligandability score;
22:  **return** The candidate binding sites and ligandability score;

---

### A.2    THE SUPPLEMENT DETAILS FOR EXPERIMENT PROCESS

| Number of Binding Sites | Number of Proteins | | |
|---|---|---|---|
| | COACH420 | Holo4k | PDBbind |
| 1 | 235 | 2442 | 5025 |
| 2 | 36 | 635 | 0 |
| 3 | 7 | 67 | 0 |
| 4 | 4 | 22 | 0 |
| >=5 | 2 | 38 | 0 |

As indicated by the data distribution presented in the table above, the majority of samples in the test dataset consist of a single binding site, with only a subset of samples containing multiple binding sites.

To address your concern, we provide more details here. Following related methods [27; 20; 37; 3; 30], the specific processing steps of our method are outlined below, and further details can be found in Section 4.1 and Appendix A.1 of our paper:

**Training Process**:

a. Following DeepSurf [37], the protein atoms within 4 Ångströms of any ligand atom are set as positive, otherwise negative.

b. EquiPocket predicts the druggability probability for each atom, using the Dice loss function for model optimization.

**Validation and Testing**:

a. In accordance with related methods [27; 20; 37; 3; 30], we define the center of a binding site as the mean position of the ligand atoms. This definition allows us to handle the proteins with multiple binding sites ($N \geq 1$).

b. We focus on predicting druggability probabilities for each atom rather than identifying specific binding site atoms.

c. Our method uses predicted atom-level probabilities in a clustering process [8] to autonomously identify ligand binding site centers and druggability, which are crucial for metrics and downstream docking tasks.

**Performance Evaluation**:

a. We evaluate based on the top-$n$ predicted binding site centers, where $n$ does not exceed the actual number of binding sites. If no predicted binding sites center is found, it is considered a failure.

b. The predicted ligand binding center with the DCC/DCA falls below a predetermined threshold, typically set to 4, is classified as successful.

Through the aforementioned process, we are able to effectively handle proteins with more than one binding site during training, validation, and testing.

## A.3 RELATED WORK

### A.3.1 BINDING SITE PREDICTION

**Computational Methods.** The computational methods for binding site prediction include geometry-based [32; 16; 55; 30; 6; 11], probe- and energy-based [28; 29; 38] and template-based [5; 50] methods: 1) Since most ligand binding sites occur on the 3D structure, geometry-based methods (POCKET [32], CriticalFinder [11], LigSite [16], Fpocket [30], etc. ) are designed to identify these hollow spaces and then rank them using the expert design geometric features. 2) Probe-based methods (SURFNET [28], Q-SiteFinder [29], etc. [12]), also known as energy-based methods, calculate the energy resulting from the interaction between protein atoms and a small-molecule probe, whose value dictates the existence of binding sites. 3)Template-based methods (FINDSITE [5], LIBRA [50], etc.) are mainly to compare the required query protein with the published protein structure database to identify the binding sites.

**Traditional Learning-based Methods.** PRANK [26] is a learning-based method that employs the traditional machine learning algorithm random forest(RF) [4]. Based on the pocket points and chemical properties from Fpocket [30] and Concavity [6], this method measures the "ligandibility" as the binding ability of a candidate pocket using the RF model. P2rank [27] is a widely used package for locating the ligand-binding pockets based on protein structures. We have studied P2Rank in the early stage of research. However, we didn't include P2Rank as a baseline mainly due to the differences in training and validation data between it and deep learning methods including DeepSite [20], Kalasanty [47], DeepSurf [37], and our EquiPocket. **Specifically, P2Rank uses data from CHEN11 and JOINED datasets, while deep learning methods commonly use scPDB. P2Rank's paper mentions that CHEN11 is more diverse than scPDB [10], affecting model performance.** However, those methods require the manual extraction of numerous features with limit upgrading.

| DCA | COACH420 | HOLO4K |
|---|---|---|
| P2Rank[protrusion] | 0.642 | 0.593 |
| P2rank | 0.683 | 0.706 |
| DeepSite | 0.564 | 0.456 |
| Kalasanty | 0.636 | 0.515 |
| deepsurf | 0.658 | 0.635 |
| EquiPocket | 0.656 | 0.662 |

The results in above table show that our method essentially matches or surpasses most deep learning methods, even outperforming P2Rank [protrusion], which uses only geometric information, and slightly trailing behind P2Rank, which benefits from a **more diverse dataset**.

**CNN-based Methods.** Over the last few years, deep learning has surpassed far more traditional ML methods in many domains. For binding site prediction task, many researchers [20; 47; 37; 21; 3] regard a protein as a 3D image, and model this task as a computer vision problem. DeepSite [20] is the first attempt to employ the CNN architecture for binding site prediction, which like P2Rank [27] treats this task as a binary classification problem and converts a protein to 3D voxelized grids. The methods FRSite [19] and Kalasanty [47] adhere to the principle of deepsite, but the former regards this task as an object detection problem, and the latter regards this task as a semantic segmentation task.

Deeppocket [3] is a method similar to p2rank, but implements a CNN-based segmentation model as the scoring function in order to more precisely locate the the binding sites. The recent CNN-based method DeepSurf [37] constructs a local 3D grid and updates the 3D-CNN architecture to mitigate the detrimental effects of protein rotation.

### A.3.2 GRAPH NEURAL NETWORKS FOR MOLECULE MODELING

There are multi-level information in molecules including atom info, chemical bonds, spatial structure, physical constraints, etc. Numerous researchers view molecules as topological structures and apply topological-based GNN models (like graph2vec [14], GAT [52], GCN [22], GCN2 [7], GIN [56] and etc. [48]) to extract the chemical info, which achieve positive outcomes. With the accumulation of structure data for molecules, spatial-based graph models (DimeNet [25] , DimeNet++ [24], SphereNet [33], SchNet [46], Egnn [44], [18], [15] and etc.) are proposed for molecule task which aggregates spatial and topological information. However, these models may not be adequate for macro-molecules due to their high calculation and resource requirements.

### A.3.3 GNN-BASED METHODS FOR POCKET TASK.

ScanNet [51]: This model constructs atom and amino acid representations based on the spatial and chemical arrangement of neighboring entities. It is trained to detect protein-protein and protein-antibody binding sites, showcasing its accuracy even with unseen protein folds. However, it should be noted that ScanNet doesn't incorporate surface geometric information of proteins, and it isn't tailored specifically for ligand-protein datasets. It utilizes a straightforward message passing approach and lacks consideration of geometric invariance. Besides, ScanNet is designed for predicting binding sites in protein and protein, protein and antibody, and protein and disordered protein interactions, making it unsuitable for ligand binding site prediction. PocketMiner [36]: This model utilizes a geometric graph model to identify cryptic pockets. Unlike our study, PocketMiner doesn't focus on pinpointing where a structure becomes a pocket, which is related to target detection or semantic tasks. Instead, its main goal is predicting the locations where cryptic pockets, already known in advance, will open—a classification prediction task. The evaluation metric used is ROC-AUC, and it is compared against molecular simulation methods. NodeCoder [2]: NodeCoder is a computational model designed for the prediction of protein residue types based on a geometric graph representation. The model encompasses six distinct residue classifications, namely ligands, peptides, ions, nucleic acid binding sites, post-translational modifications, and transmembrane regions. It is crucial to emphasize that NodeCoder primarily serves as a residue classification tool rather than a protein pocket detection algorithm. PIPGCN [13]: This model employs GNN model to aggregate information from different protein residues and predict their categories in the Docking Benchmark Dataset. These categories include residues that interact with ligands and those that do not. It's crucial to emphasize that PIPGCN is designed for a classification task rather than target detection or semantic segmentation.

### A.4 EXPERIMENT DETAILS

### A.4.1 DATASET

scPDB [10] is the famous dataset for binding site prediction, which contains the protein structure, ligand structure, and 3D cavity structure generated by VolSite [9]. The 2017 release of scPDB is used for training and cross-validation of our framework, which contains 17,594 structures, 16,034 entries, 4,782 proteins, and 6,326 ligands. PDBbind [54] is a well-known and commonly used dataset for the research of protein-ligand complex. It contains the 3D structures of proteins, ligands, binding sites, and accurate binding affinity results determined in the laboratory. We use the release of v2020,

Table 2: Summary of Dataset

| DataSet | Average | | | |
|---------|---------|---------------|----------------|--------------|
| | Atom Num | Atom in Surface | Surface Points | Target Atoms |
| scPDB | 4205 | 2317 | 24010 | 47 |
| COACH420 | 2123 | 1217 | 12325 | 58 |
| HOLO4k | 3845 | 2052 | 20023 | 106 |
| PDBbind | 3104 | 1677 | 17357 | 37 |

which consists of two parts: general set (14, 127 complexes) and refined set (5,316 complexes). The general set contains all protein-ligand complexes. The refined set contains better-quality compounds selected from the general set, which is used for the test in our experiments. COACH 420 and HOLO4K are two test datasets for the binding site prediction, which are first introduced by [27]. Consistent with [27; 37; 3], we use the mlig subsets of each dataset for evaluation, which contain the relevant ligands for binding site prediction.

## A.5 DATA PREPARATION

We perform the following four processing steps: i) Cluster the structures in scPDB by their Uniprot IDs, and select the longest sequenced protein structures from every cluster as the train data [21]. Finally, 5,372 structures are selected out. ii) Split proteins and ligands for the structures in COACH420 and HOLO4k, according to the research [27] . iii) Clean protein by removing the solvent, hydrogens atoms. Using MSMS [43] to generate the solvent-accessible surface of a protein. iv) Read the protein file by RDKIT [49], and extract the atom and chemical bond features. Remove the error structures.

### A.5.1 EVALUATION METRICS

**DCC** is the distance between the predicted binding site center and the true binding site center. **DCA** is the shortest distance between the predicted binding site center and any atom of the ligand. The samples with DCC(DCA) less than the threshold are considered successful. The samples without any binding site center are considered failures. Consistent with [20; 37; 3; 47], threshold is set to 4 Å. We use **Success Rate** and **Failure Rate** to evaluate experimental performance.

$$\text{Success Rate(DCC/DCA)} = \frac{1(\{\text{Predicted sites}|\text{DCC/DCA} < \text{threshold}\})}{1(\{\text{True sites}\})},$$
$$\text{Failure Rate} = \frac{1(\{\text{Protein}|1(\text{predicted binding center}) = 0\})}{1(\{\text{Protein}\})},$$

(11)

where $1(\cdot)$ represents the cardinality of a set. After ranking the predicted binding sites, we take the same number with the true binding sites to calculate the success rate.

### A.5.2 BINDING SITES CENTER

The CNN-based methods [20; 3; 47] consider a protein as a 3D image, convert it to a voxel representation by discretizing it into grids and calculate the geometric center of binding site $center_{cnn}$ according to the grid of the cavity or ligand. They label the **grid** as positive if its geometric center is closer than 4Å to the binding sites geometric center. Therefore, the prediction objects of these models actually contain the grid of **ligand atoms**. The predicted binding site center $\hat{center}_{cnn}$ of CNN-based methods is calculated according to the positive grid. For our EquiPocket, we label the **protein atoms** within 4Å of any ligand atom as positive and negative otherwise. Therefore, there is a natural gap in the prediction object between our framework and CNN-based methods, which also lead to the natural gap for the center of predicted binding site. In order to reduce the metric difference caused by the different prediction objects, we get the predicted binding site center $\hat{center}_{equipocket}$ as follow: Ww use $pos_i \in \mathbb{R}^3$ to represent the position of protein atom $v_i$, $center_i \in \mathbb{R}^3$ to represent the nearest surface point center, $\hat{pos}_i^L \in \mathbb{R}^3$ to represent the predicted position of nearest ligand atom from the protein atom $v_i$. The $\hat{pos}_i^L$ is used to calculate the geometric center of binding site.

$$\widehat{pos}_i^L = pos_i + threshold \cdot \frac{(center_i - pos_i)}{|center_i - pos_i|},$$

(12)

Where $threshold4$ is set to 4, because we label the protein atoms within **4Å** of any ligand atom as positive and negative otherwise.

### A.5.3 CROSS-VALIDATION

We shuffled the training data and divided the data into 5 parts, taking one of them at a time as the validation set. We use 5-fold cross-validation and report the mean and standard deviation.

### A.5.4 ENVIRONMENT AND PARAMETER SETTINGS

For geometric-based method Fpocket, we use its published tool. For CNN-based methods kalasanty and DeepSurf, we use their published pre-train models. For GNN-based models, the number of layers is set to 3 except GAT. For GAT, we set the number to 1. For GIN, we set the initial $\epsilon$ to 0 and make it trainable. For GCN2, we set the strength of the initial residual connection $\alpha$ to 0.5 and the strength of the identity mapping $\beta$ to 1. For SchNet, EGNN, DimeNet++, SphereNet as baseline models, we set the cutoff distance to 5. For our EquiPocket, we use Adam optimizer for model training with a learning rate of 0.0001 and set the batch size as 8. The basic dimensions of node and edge embeddings are both set to 128. The dropout rate is set to 0.1. The probe radius in MSMS to generate solvent-accessible surface of a protein is set to 1.5. We implement our EquiPocket framework in PyTorch Geometric, all the experiments are conducted on a machine with an NVIDIA A100 GPU (80GB memory). We take 5-fold cross validation on training data scPDB and use valid loss to save checkpoint.

### A.5.5 BASELINE CODES

The result of DeepSite comes from [37], because they did not provide a pre-train model. Table 3 describes sources of baseline codes.

Table 3: Sources of baseline codes and pre-train models.

| Methods | URL |
|---|---|
| Fpocker | https://github.com/Discngine/fpocket |
| kalasanty | https://gitlab.com/cheminfIBB/kalasanty |
| DeepSurf | https://github.com/stemylonas/DeepSurf |
| GAT | https://github.com/pyg-team/pytorch_geometric |
| GCN | https://github.com/pyg-team/pytorch_geometric |
| GCN2 | https://github.com/chennnM/GCNII |
| SchNet | https://github.com/pyg-team/pytorch_geometric |
| DimeNet++ | https://github.com/pyg-team/pytorch_geometric |
| EGNN | https://github.com/vgsatorras/egnn/ |

### A.5.6 PROBE RADIUS

To evaluate the sensitivities of this parameter, we implement various radius of probe (1, 1.5, 2) and provide the number of surface points, GPU memory consumption, failure rate, and the DCC accuracy of our EquiPocket on the test sets.

Table 4: Experimental results with different probe radius by MSMS.

| Radius | Surface Points | GPU | Failure Rate | COACH420 | HOLO4K | PDBbind2020 |
|---|---|---|---|---|---|---|
| 1 | 28030 | 44 | 0.053 | 0.433(0.018) | 0.338(0.008) | 0.549(0.005) |
| 1.5 | 24010 | 33 | 0.051 | 0.423(0.014) | 0.337(0.006) | 0.545(0.010) |
| 2 | 21725 | 28 | 0.096 | 0.393(0.024) | 0.289(0.004) | 0.524(0.012) |

### A.5.7 CASE STUDY

We also display two examples of our EquiPocket and other methods in Figure 8. We take two proteins, 1f8e (with 12,268 atoms) and 5ei3 (with 1,572 atoms), from the test dataset PDBbind. As

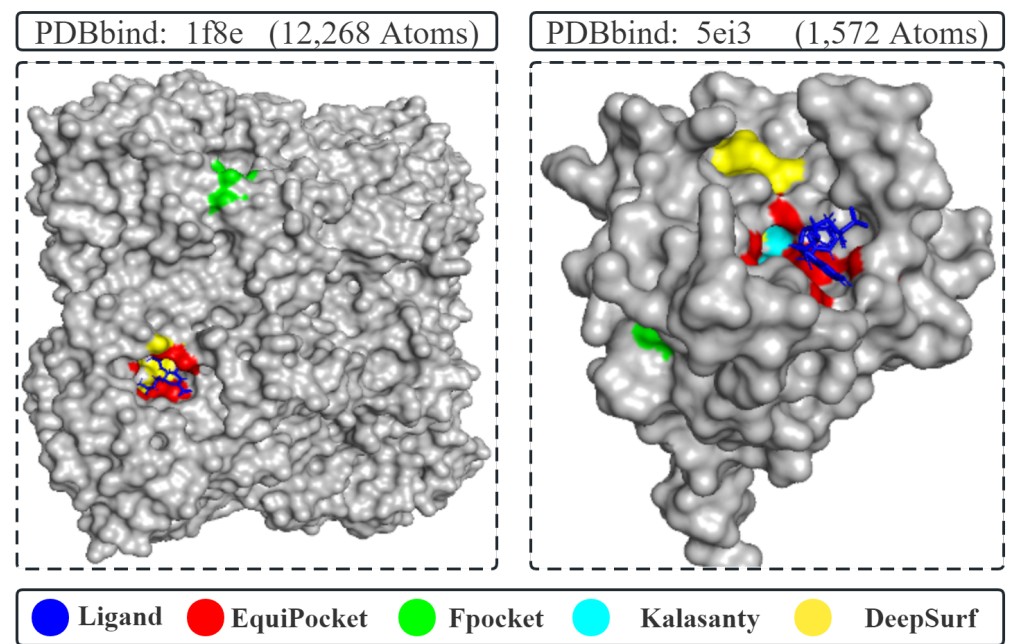

Figure 8: Case Study.

can be seen from Figure 8: The binding sites predicted by the geometry-based method Fpocket are extremely distant from the actual binding sites. This is due to the fact that this method prioritizes local geometric information and disregards the multi-level structure information of proteins, resulting in limited scope and weak performance. The CNN-based method Kalasanty did not provide any predicted binding site for protein 1f8e. We conjecture that this method restricts the protein within a specific space size which is highly susceptible to failure with large proteins. The recently-proposed CNN-based method DeepSurf takes local grids on the protein surface, which can address the issue of fixed space size. However, the prediction of binding sites in protein 5ei3 by DeepSurf is far from the ground truth because the CNN-based methods are defective in obtaining geometric and chemical features. Our EquiPocket framework is unaffected by the shortcomings of the aforementioned methods, allowing it to achieve superior outcomes for both large and small proteins.

## A.6 THE INFERENCE SPEED OF DIFFERENT METHODS

The comparison of various methods for predicting 100 proteins reveals the following:

Table 5: The inference speed of different methods.

| Method | Type | Time (s) per 100 proteins | Average DCC |
|---|---|---|---|
| fpocket | Geometric-based | 23 | 0.214 |
| Kalasanty | 3D-CNN | 86 | 0.321 |
| DeepSurf | | 641 | 0.366 |
| EquiPocket | Ours | 37 | 0.431 |

Table 6: The summary of our test dataset.

| dataset | Average Atom Num | Average Atom in Surface | Average True binding sites |
|---|---|---|---|
| COACH420 | 2123 | 1217 | 1.2 |
| HOLO4K | 3845 | 2052 | 2.4 |
| PDBbind | 3104 | 1677 | 1 |

fpocket[5]: Fastest with only 23 seconds for 100 proteins, leveraging manually defined geometric features. However, its performance metrics are not notable.

Kalasanty[6] and DeepSurf[7]: Both are 3D-CNN-based. DeepSurf, using detailed local grids on protein surfaces, outperforms Kalasanty in metrics but is slower and the least efficient among the methods compared.

EquiPocket: Our method takes 47 seconds per 100 proteins and shows the best DCC metrics. It's faster than 3D-CNN methods but slower than geometric-based ones. This is due to 3D-CNN methods transforming proteins into 3D images (eg. 36 * 36 * 36 grids [6, 7, 8]), increasing computational costs compared to our using atom information (averages 2000-3000 nodes in a protein). EquiPocket also integrates surface features with Surface-EGNN, enhancing efficiency over DeepSurf.

## A.7 THE INFORMATIVE ABLATION EXPERIMENT FOR OUR TWO FEATURE EXTRACTORS

Table 7: The informative ablation experiment for our two feature extractors.

|  |  | COACH420 |  | HOLO4k |  | PDBbind |  |
| --- | --- | --- | --- | --- | --- | --- | --- |
|  | Fail Ratio | DCC | DCA | DCC | DCA | DCC | DCA |
| EquiPocket/L | 0.13 | 0.355 | 0.546 | 0.296 | 0.574 | 0.465 | 0.606 |
| EquiPocket/R | 0.09 | 0.364 | 0.541 | 0.294 | 0.598 | 0.474 | 0.627 |
| EquiPocket/LR | 0.16 | 0.308 | 0.502 | 0.268 | 0.543 | 0.409 | 0.566 |
| EquiPocket | 0.05 | 0.423 | 0.656 | 0.337 | 0.662 | 0.545 | 0.721 |

Our model predominantly comprises two feature extractors: local geometric modeling module and global structural modeling module, subsequently followed by the surface-EGNN model. In response to your valuable suggestion, we propose the following definitions:

EquiPocket/L: This variant of EquiPocket excludes local geometric modeling module.

EquiPocket/R: This variant of EquiPocket excludes global structural modeling module.

EquiPocket/LR: This variant of EquiPocket excludes both local geometric modeling module and global structural modeling module.

Analysis shows that omitting any of these modules negatively impacts performance. Specifically, excluding either the local geometric (L) or global structural (R) module leads to a 10%-15% decrease in DCC/DCA metrics; removing both L and R modules results in a more significant drop of 20%-25%. These results highlight the essential role of both feature extractors in predicting ligand binding sites. Notably, the more pronounced performance decline when omitting the local geometric module (L) suggests its higher importance in protein pocket prediction. This finding is consistent with current trends where methods like Fpocket[5], P2rank[4], and DeepSurf[7] primarily utilize geometric features for binding site prediction.

## A.8 THE DETAILED EXPERIMENT RESULTS OF DENSE ATTENTION

Table 8: The detailed experiment results of Dense attention.

| Atom Num | Protein Num | Ratio | Cumsum Ratio | DCC of EquiPocket(w/o attention) | DCC of EquiPocket |
| --- | --- | --- | --- | --- | --- |
| 0-1000 | 296 | 0.04 | 0.04 | 0.328 | 0.428 |
| 1000-2000 | 2193 | 0.27 | 0.3 | 0.547 | 0.621 |
| 2000-3000 | 2534 | 0.31 | 0.61 | 0.551 | 0.590 |
| 3000-4000 | 1143 | 0.14 | 0.75 | 0.343 | 0.388 |
| 4000-5000 | 619 | 0.08 | 0.82 | 0.261 | 0.255 |
| >=5000 | 1440 | 0.18 | 1 | 0.161 | 0.153 |

Our paper's Figure 6C compares EquiPocket with/without the Dense Attention module. We removed the Attention Module in EquiPocket (w/o attention) for this comparison. The above table presents results across different protein sizes: It is evident that Dense Attention notably improves prediction for proteins with less than 4000 nodes. However, for larger proteins, exceeding 4000

nodes, there's no significant performance difference. These findings highlight that Dense Attention boosts predictive accuracy for smaller proteins while maintaining performance for larger ones.

## A.9 THE DETAILED EXPERIMENT RESULTS OF RELATIVE DIRECTION

The core reasons are as follows: the relative direction between a protein atom and its nearest ligand atom effectively captures the local geometric information of the binding sites on a protein. Different from the previous work [27; 20; 37], our method can output E(3)-equivariant coordinates (detailed in Section 3.3 of our paper). To better capture the geometric details of the protein surface, we introduced more detailed relative direction as a supplementary target.

Table 9: The detailed experiment results of relative direction.

| Atom_num | Protein num | Ratio | Cumsum ratio | DCC of EquiPocket(w/o Direction loss) | DCC of EquiPocket |
|----------|-------------|-------|--------------|---------------------------------------|-------------------|
| 0-1000 | 296 | 0.04 | 0.04 | 0.319 | 0.428 |
| 1000-2000 | 2193 | 0.27 | 0.3 | 0.587 | 0.621 |
| 2000-3000 | 2534 | 0.31 | 0.61 | 0.551 | 0.590 |
| 3000-4000 | 1143 | 0.14 | 0.75 | 0.371 | 0.388 |
| 4000-5000 | 619 | 0.08 | 0.82 | 0.258 | 0.255 |
| >=5000 | 1440 | 0.18 | 1 | 0.144 | 0.153 |

The corresponding ablation results have been shown in Figure 6 (c) of our original paper, and detailed results are presented below. EquiPocket (w/o Direction Loss) represents our EquiPocket method removing the relative direction prediction module. It can be observed that when the relative direction prediction module is removed, our method's performance drops for proteins of different sizes. This is especially notable for proteins with fewer than 3000 atoms, which account for 60% of the samples. If the relative direction prediction is removed, the performance drops by approximately 10%. These results demonstrate the effectiveness of our designed relative direction target.

## A.10 THE COMPARISON RESULTS FOR EQ.3

In Eq.3, $g_i$ is the geometric embedding for a protein atom, learned from surrounding surface probes, and $s_i$ denotes the local geometric properties of these probes with properties such as distances and angle to protein atoms, the surface center, neighboring probes, and so on. Initially, we apply an MLP to these features, followed by pooling. This process transforms the geometric properties before aggregating them into the protein node's geometric embedding. However, since each property of $s_i$ itself carries meaningful information. We are concerned that applying MLP first and then pooling might weaken the transmission of this information. Therefore, we take the second part of the equation.

Table 10: The comparison results for Eq.3

| Model | Fail Ratio | Coach429 | | HOLO4k | | PDBbind | |
|-------|------------|----------|------|--------|------|---------|------|
| | | DCC | DCA | DCC | DCA | DCC | DCA |
| EquiPocket-former | 0.16 | 0.389 | 0.606 | 0.330 | 0.637 | 0.507 | 0.660 |
| EquiPocket-latter | 0.16 | 0.407 | 0.617 | 0.319 | 0.644 | 0.529 | 0.676 |
| EquiPocket | 0.05 | 0.423 | 0.656 | 0.337 | 0.662 | 0.545 | 0.721 |

To highlight the effectiveness of the features in Eq.3, we carried out extra experiments with "EquiPocket-former" focusing on the equation's initial part and "EquiPocket-latter" on its latter part. The findings show: Using either feature alone diminishes the predictive performance compared to the full EquiPocket model. Specifically, "EquiPocket-former" alone sees about a 10% drop, while "EquiPocket-latter" alone results in around a 5% reduction. This outcome underscores the necessity of both features, with the latter part having a more substantial impact on our model's performance.

### A.10.1 THE 5-FOLD RESULTS FOR EQUIPOCKET

Table 11: The 5-fold results for EquiPocket.

| Methods | Fold | Param (M) | failure Rate ↓ | COACH420 DCC↑ | COACH420 DCA↑ | HOLO4K DCC↑ | HOLO4K DCA↑ | PDBbind2020 DCC↑ | PDBbind2020 DCA↑ |
|---|---|---|---|---|---|---|---|---|---|
| EquiPocket-L | 0 | 0.15 | 0.598 | 0.083 | 0.160 | 0.038 | 0.128 | 0.049 | 0.124 |
| EquiPocket-L | 1 | 0.15 | 0.557 | 0.064 | 0.165 | 0.046 | 0.138 | 0.055 | 0.142 |
| EquiPocket-L | 2 | 0.15 | 0.571 | 0.074 | 0.177 | 0.045 | 0.139 | 0.052 | 0.122 |
| EquiPocket-L | 3 | 0.15 | 0.462 | 0.059 | 0.173 | 0.042 | 0.138 | 0.052 | 0.129 |
| EquiPocket-L | 4 | 0.15 | 0.472 | 0.072 | 0.180 | 0.048 | 0.146 | 0.049 | 0.143 |
| EquiPocket-G | 0 | 0.42 | 0.305 | 0.135 | 0.330 | 0.122 | 0.400 | 0.142 | 0.302 |
| EquiPocket-G | 1 | 0.42 | 0.291 | 0.175 | 0.385 | 0.128 | 0.405 | 0.145 | 0.302 |
| EquiPocket-G | 2 | 0.42 | 0.295 | 0.145 | 0.357 | 0.121 | 0.407 | 0.145 | 0.305 |
| EquiPocket-G | 3 | 0.42 | 0.278 | 0.169 | 0.367 | 0.127 | 0.406 | 0.133 | 0.292 |
| EquiPocket-G | 4 | 0.42 | 0.292 | 0.152 | 0.367 | 0.133 | 0.411 | 0.151 | 0.308 |
| EquiPocket-LG | 0 | 0.50 | 0.235 | 0.225 | 0.442 | 0.183 | 0.498 | 0.273 | 0.463 |
| EquiPocket-LG | 1 | 0.50 | 0.207 | 0.220 | 0.460 | 0.189 | 0.509 | 0.280 | 0.468 |
| EquiPocket-LG | 2 | 0.50 | 0.203 | 0.184 | 0.440 | 0.180 | 0.510 | 0.269 | 0.459 |
| EquiPocket-LG | 3 | 0.50 | 0.224 | 0.215 | 0.448 | 0.186 | 0.500 | 0.275 | 0.465 |
| EquiPocket-LG | 4 | 0.50 | 0.231 | 0.213 | 0.431 | 0.179 | 0.492 | 0.272 | 0.456 |
| EquiPocket | 0 | 1.70 | 0.054 | 0.423 | 0.656 | 0.341 | 0.665 | 0.558 | 0.715 |
| EquiPocket | 1 | 1.70 | 0.053 | 0.431 | 0.660 | 0.329 | 0.668 | 0.538 | 0.725 |
| EquiPocket | 2 | 1.70 | 0.041 | 0.443 | 0.664 | 0.336 | 0.660 | 0.550 | 0.724 |
| EquiPocket | 3 | 1.70 | 0.051 | 0.411 | 0.646 | 0.338 | 0.668 | 0.532 | 0.723 |
| EquiPocket | 4 | 1.70 | 0.053 | 0.407 | 0.654 | 0.345 | 0.652 | 0.546 | 0.719 |

