# OpenReview forum: "EquiPocket: an E(3)-Equivariant Geometric Graph Neural Network for Ligand Binding Site Prediction"
_ICLR.cc/2024/Conference — Submitted to ICLR 2024_

### Official Review · Reviewer_fzz6 · 2023-10-31

**Soundness:** 2 fair
**Presentation:** 3 good
**Contribution:** 2 fair
**Rating:** 5
**Confidence:** 4

**Summary:**

The paper introduces an e(3)-equivariant geometric graph neural network for ligand binding site prediction. This model consists of three modules including the local/global geometric modeling module and surface message passing module, which are used to address several issues of the previous approach. The author compares their method with other methods and shows it has the best performance in terms of model parameters efficiency and accuracy.

**Strengths:**

1. The writing is generally great and the description of the proposed method is clear.
2. The discussion regarding the protein size shift is great.

**Weaknesses:**

1. Geometric aware E(3)-equivariant GNN has been applied to several very related tasks like protein-ligand docking [1]. Consequently, the novelty of introducing E(3)-equivariant GNNs may be diminished.
2. I think it might be also necessary to compare with protein-ligand docking methods as binding site prediction is one of their outputs.


[1]: Zhang, Yangtian, et al. "E3bind: An end-to-end equivariant network for protein-ligand docking." arXiv preprint arXiv:2210.06069 (2022).

**Questions:**

1. It would be better to also compare the inference speed of different methods. Because this approach does massage passing on full protein atoms graph, it appears the inference speed would be very slow.
2. The setting for some baseline methods (EGNN, SchNet) needs to be more clear. For example, what kinds of graphs do you use for the EGNN? Do you also use the surface atom graph?
3. A more informative ablation would involve comparing EquiPocket, EquiPocket/L, EquiPocket/R, and EquiPocket/LR, where the symbol '/' means exclude. This is because L and R appear to be two feature extractors.

---

> ### Author Response · Authors · 2023-11-19
> **Part 1/2 of Response to Reviewer fzz6**
>
> Thank you for your valuable feedback! To address your concerns, we have included additional explanations as follows.
>
> >__W1: Geometric aware E(3)-equivariant GNN has been applied to several very related tasks like protein-ligand docking [1]. Consequently, the novelty of introducing E(3)-equivariant GNNs may be diminished.__
>
> Thanks for the comment. There could be a misunderstanding regarding our research task. While it's true that E(3)-equivariant GNNs are used in tasks such as protein-ligand docking (like E3bind [1]), our EquiPocket focuses on protein pocket prediction, a distinct task from docking.
>
> Docking methods like E3Bind [1] and TankBind [2] predict protein-ligand binding poses and affinities, often using method like P2rank [4] to first identify candidate binding sites (which is also called protein pocket in our context).
> **Our EquiPocket and other related works [5, 6, 7, 8, 9] share the same goal as P2rank, identifying the center of candidate binding sites on the protein rather than locating the exact interface between each ligand and the target protein. The area around this center is seen as protein pocket for downstream tasks such as protein-ligand docking that will specify more precise protein-ligand interface around the predicted protein pocket.** We have studied P2Rank[4] in our paper (Section A3.1).
>
> Most current deep learning methods [6, 7, 8] for protein pocket prediction use CNNs and require voxelization of protein structures.
> EquiPocket innovatively uses a geometric-aware E(3)-equivariant GNN, which is a unique model unexplored by other research in this field.
> As also noted by Reviewer 9vNs, "the idea of using an E(3)-equivariant GNN for binding site prediction is of interest especially in the bioinformatic domain."
>
>
> >__W2: I think it might be also necessary to compare with protein-ligand docking methods as binding site prediction is one of their outputs.__
>
> Thank your for the comment. Again, we highlight that protein-ligand docking and protein pocket prediction are two different tasks. A general pipeline in current methods (E3Bind [1] and TankBind [2]) is that we first predict the protein pocket, and then conduct protein-ligand docking and infer how specific ligands interact with the atoms around the predicted pocket. Although the protein-ligand docking methods also output binding site, this area is a fine-grained subset of the atoms around the protein pocket. Hence, it is not reasonable to  compare with protein-ligand docking methods. We have further introduced the difference between these two tasks in the revision.
>
>
> >__Q1: It would be better to also compare the inference speed of different methods. Because this approach does massage passing on full protein atoms graph, it appears the inference speed would be very slow.__
>
> Thank you for the feedback. The comparison of various methods for predicting 100 proteins reveals the following:
>
> |   Method   |       Type      | Time (s)  per 100 proteins | Average DCC |
> |:----------:|:---------------:|:--------------------------:|:-----------:|
> |   fpocket  | Geometric-based |             23             |    0.214    |
> |  Kalasanty | 3D-CNN          |             86             |    0.321    |
> |  DeepSurf  | 3D-CNN          |             641            |    0.366    |
> | EquiPocket |       Ours      |             37             |    0.431    |
>
>
> | Dataset  | Average  Atom Num | Average  Atom in Surface | Average the True Center of Binding Sites  |
> |----------|-------------------|--------------------------|-------------------------------------------|
> | COACH420 | 2123              | 1217                     | 1.2                                       |
> | HOLO4K   | 3845              | 2052                     | 2.4                                       |
> | PDBbind  | 3104              | 1677                     | 1                                         |
>
>
>
> fpocket[5]: Fastest with only 23 seconds for 100 proteins, leveraging manually defined geometric features. However, its performance metrics are not notable.
>
> Kalasanty[6] and DeepSurf[7]: Both are 3D-CNN-based. DeepSurf, using detailed local grids on protein surfaces, outperforms Kalasanty in metrics but is slower and the least efficient among the methods compared.
>
> EquiPocket: Our method takes 47 seconds per 100 proteins and shows the best DCC metrics. It's faster than 3D-CNN methods but slower than geometric-based ones. This is due to 3D-CNN methods transforming proteins into 3D images (eg. 36 * 36 * 36 grids~[6, 7, 8]), increasing computational costs compared to our using atom information (average 2000-3000 nodes in a protein). EquiPocket also integrates surface features with Surface-EGNN, enhancing efficiency over DeepSurf.
>
> We have included these results and analyses in the revised paper. Thank you for your nice suggestion.

---

> ### Author Response · Authors · 2023-11-19
> **Part 2/2 of Response to Reviewer fzz6**
>
> >__Q2: The setting for some baseline methods (EGNN, SchNet) needs to be more clear. For example, what kinds of graphs do you use for the EGNN? Do you also use the surface atom graph?__
>
> We focused on using EGNN[10] and SchNet[11] solely on the protein structure graph for two main reasons:
>
> Both EGNN and SchNet were originally developed for representing molecular nodes and structures, making them ideal to test whether solely taking protein structural information is sufficient for predicting ligand binding sites.
>
> Our method EquiPocket differs significantly from previous prediction method [4, 5, 6, 7, 8] by employing surface atom graph, which represents an innovative contribution and differentiates from the conventional protein structure graph as baseline models.
>
> >__Q3: A more informative ablation would involve comparing EquiPocket, EquiPocket/L, EquiPocket/R, and EquiPocket/LR, where the symbol '/' means exclude. This is because L and R appear to be two feature extractors.__
>
> Our model predominantly comprises two feature extractors: local geometric modeling module and global structural modeling module, subsequently followed by the surface-EGNN model. In response to your valuable suggestion, we propose the following definitions:
>
> EquiPocket/L: This variant of EquiPocket excludes local geometric modeling module.
>
> EquiPocket/R: This variant of EquiPocket excludes global structural modeling module.
>
> EquiPocket/LR: This variant of EquiPocket excludes both local geometric modeling module and global structural modeling module.
>
> | Dataset       |            | COACH420 |        | HOLO4k |        | PDBbind |        |
> |---------------|------------|----------|--------|--------|--------|---------|--------|
> | Model         | Fail Ratio | DCC      | DCA    | DCC    | DCA    | DCC     | DCA    |
> | EquiPocket/L  | 0.13       | 0.355    | 0.546  | 0.296  | 0.574  | 0.465   | 0.606  |
> | EquiPocket/R  | 0.09       | 0.364    | 0.541  | 0.294  | 0.598  | 0.474   | 0.627  |
> | EquiPocket/LR | 0.16       | 0.308    | 0.502  | 0.268  | 0.543  | 0.409   | 0.566  |
> | EquiPocket    | 0.05       | 0.423    | 0.656  | 0.337  | 0.662  | 0.545   | 0.721  |
>
> Analysis shows that omitting any of these modules negatively impacts performance. Specifically, excluding either the local geometric (L) or global structural (R) module leads to a 10\%-15\% decrease in DCC/DCA metrics; removing both L and R modules results in a more significant drop of 20\%-25\%.
> These results highlight the essential role of both feature extractors in predicting ligand binding sites. Notably, the more pronounced performance decline when omitting the local geometric module (L) suggests its higher importance in protein pocket prediction. This finding is consistent with current trends where methods like Fpocket[5], P2rank[4], and DeepSurf[7] primarily utilize geometric features for binding site prediction.
>
>
> **Reference**
>
> [1] Zhang Y, Cai H, Shi C, et al. E3bind: An end-to-end equivariant network for protein-ligand docking[J]. arXiv preprint arXiv:2210.06069, 2022.
>
> [2] Lu W, Wu Q, Zhang J, et al. Tankbind: Trigonometry-aware neural networks for drug-protein binding structure prediction[J]. Advances in neural information processing systems, 2022, 35: 7236-7249.
>
> [3] Liao Z, You R, Huang X, et al. DeepDock: enhancing ligand-protein interaction prediction by a combination of ligand and structure information[C]//2019 IEEE International Conference on Bioinformatics and Biomedicine (BIBM). IEEE, 2019: 311-317.
>
> [4] Krivák R, Hoksza D. P2Rank: machine learning based tool for rapid and accurate prediction of ligand binding sites from protein structure[J]. Journal of cheminformatics, 2018, 10: 1-12.
>
> [5] Le Guilloux V, Schmidtke P, Tuffery P. Fpocket: an open source platform for ligand pocket detection[J]. BMC bioinformatics, 2009, 10(1): 1-11.
>
> [6] Stepniewska-Dziubinska M M, Zielenkiewicz P, Siedlecki P. Improving detection of protein-ligand binding sites with 3D segmentation[J]. Scientific reports, 2020, 10(1): 5035.
>
> [7] Mylonas S K, Axenopoulos A, Daras P. DeepSurf: a surface-based deep learning approach for the prediction of ligand binding sites on proteins[J]. Bioinformatics, 2021, 37(12): 1681-1690
>
> [8] Jiménez J, Doerr S, Martínez-Rosell G, et al. DeepSite: protein-binding site predictor using 3D-convolutional neural networks[J]. Bioinformatics, 2017, 33(19): 3036-3042.
>
> [9] Tubiana J, Schneidman-Duhovny D, Wolfson H J. Scannet: A web server for structure-based prediction of protein binding sites with geometric deep learning[J]. Journal of Molecular Biology, 2022, 434(19): 167758.
>
> [10] Satorras V G, Hoogeboom E, Welling M. E (n) equivariant graph neural networks[C]//International conference on machine learning. PMLR, 2021: 9323-9332.
>
> [11] Schütt K T, Sauceda H E, Kindermans P J, et al. Schnet–a deep learning architecture for molecules and materials[J]. The Journal of Chemical Physics, 2018, 148(24).

---

> ### Author Response · Authors · 2023-11-22
> **Rebuttal Deadline Reminder**
>
> Dear reviewer fzz6:
>
> Thank you very much for your review.
>
> I would like to kindly remind you that the rebuttal period is coming to an end. Could you please inform us if our responses have resolved your concerns, or if there are any other questions you need us to address?

---

> > ### Comment · Reviewer_fzz6 · 2023-11-23
> >
> > Thanks for providing additional details! They are quite valuable.
> >
> > Thanks for pointing out that E3Bind results would be close to P2Rank. However, I've noticed that the revised manuscript lacks a direct comparison between your approach and P2Rank. I think such a comparison is important, especially considering that P2Rank has demonstrated a significant performance advantage over one of your baseline methods, DeepSite. So, I will keep my score.

---

> > > ### Author Response · Authors · 2023-11-23
> > > **Urgent! New Rebuttal Response Reminder**
> > >
> > > Dear reviewers and bros fzz6:
> > >
> > > We have responded to your concerns and updated the revised paper.
> > >
> > > May I ask if this has resolved your additional concern.
> > >
> > > Thanks a lot.

---

> > > > ### Comment · Reviewer_fzz6 · 2023-11-23
> > > >
> > > > Thanks for adding these results!
> > > >
> > > > It appears to me that this result is not robust enough to show your method is better than P2Rank as its performance is, in fact, lower than P2Rank. While I acknowledge that P2Rank may derive benefits from its more diverse training dataset, quantifying this advantage is less feasible. So I think maybe you need to either train your model on their dataset or re-train their model on your dataset.
> > > >
> > > > It is great to know that your model is better than P2Rank (protrusion), indicating a potential superior ability to capture geometric information compared to P2Rank. However, it is not intuitively clear to me whether the additional information captured by your model isn't already accounted for by the other features used in P2Rank.
> > > >
> > > > Thanks for your hard work! As the results don't actually address my concern, I will maintain the current score.

---

> > > > > ### Author Response · Authors · 2023-11-23
> > > > > **Response to Reviewer fzz6**
> > > > >
> > > > > Dear reviewers and bros fzz6:
> > > > >
> > > > > Thank you for your reply.
> > > > >
> > > > > I am somewhat heart-broken by this result, mainly because almost all deep learning models for ligand binding site prediction in the experiments did not compare with P2rank, affected by their data differences.
> > > > >
> > > > > Nonetheless, I greatly appreciate your reply. Thanks a lot.

---

> ### Author Response · Authors · 2023-11-23
> **Response to Reviewer fzz6**
>
> Dear Reviewer fzz6:
>
> Thank you very much for your reply.
>
> We have studied P2Rank in our paper (Section A3.1).  P2Rank[4] has great differences in training and validation data with deep learning methods including DeepSite[8], Kalasanty[7], DeepSurf[6],  and our EquiPocket. **Specifically, P2Rank uses data from CHEN11 and JOINED datasets, while deep learning methods commonly use scPDB[12]. P2Rank's paper mentions that CHEN11 is more diverse than scPDB**, affecting model performance.The comparison results are as following:
>
> | DCA                | COACH420 | HOLO4K |
> |--------------------|----------|--------|
> | P2Rank[protrusion] | 0.642    | 0.593  |
> | P2rank             | 0.683    | 0.706  |
> | DeepSite           | 0.564    | 0.456  |
> | Kalasanty          | 0.636    | 0.515  |
> | deepsurf           | 0.658    | 0.635  |
> | EquiPocket         | 0.656    | 0.662  |
>
> The results in above table show that our method essentially matches or surpasses most deep learning methods, even outperforming P2Rank [protrusion], which uses only geometric information, and slightly trailing behind P2Rank, which benefits from a more diverse dataset.
> **These results has been added to the revised paper.**
>
> Could you please inform us if this response have resolved your concerns, or if there are any other questions you need us to address?
>
> **Reference**
>
> [1] Zhang Y, Cai H, Shi C, et al. E3bind: An end-to-end equivariant network for protein-ligand docking[J]. arXiv preprint arXiv:2210.06069, 2022.
>
> [2] Lu W, Wu Q, Zhang J, et al. Tankbind: Trigonometry-aware neural networks for drug-protein binding structure prediction[J]. Advances in neural information processing systems, 2022, 35: 7236-7249.
>
> [3] Liao Z, You R, Huang X, et al. DeepDock: enhancing ligand-protein interaction prediction by a combination of ligand and structure information[C]//2019 IEEE International Conference on Bioinformatics and Biomedicine (BIBM). IEEE, 2019: 311-317.
>
> [4] Krivák R, Hoksza D. P2Rank: machine learning based tool for rapid and accurate prediction of ligand binding sites from protein structure[J]. Journal of cheminformatics, 2018, 10: 1-12.
>
> [5] Le Guilloux V, Schmidtke P, Tuffery P. Fpocket: an open source platform for ligand pocket detection[J]. BMC bioinformatics, 2009, 10(1): 1-11.
>
> [6] Stepniewska-Dziubinska M M, Zielenkiewicz P, Siedlecki P. Improving detection of protein-ligand binding sites with 3D segmentation[J]. Scientific reports, 2020, 10(1): 5035.
>
> [7] Mylonas S K, Axenopoulos A, Daras P. DeepSurf: a surface-based deep learning approach for the prediction of ligand binding sites on proteins[J]. Bioinformatics, 2021, 37(12): 1681-1690
>
> [8] Jiménez J, Doerr S, Martínez-Rosell G, et al. DeepSite: protein-binding site predictor using 3D-convolutional neural networks[J]. Bioinformatics, 2017, 33(19): 3036-3042.
>
> [9] Tubiana J, Schneidman-Duhovny D, Wolfson H J. Scannet: A web server for structure-based prediction of protein binding sites with geometric deep learning[J]. Journal of Molecular Biology, 2022, 434(19): 167758.
>
> [10] Satorras V G, Hoogeboom E, Welling M. E (n) equivariant graph neural networks[C]//International conference on machine learning. PMLR, 2021: 9323-9332.
>
> [11] Schütt K T, Sauceda H E, Kindermans P J, et al. Schnet–a deep learning architecture for molecules and materials[J]. The Journal of Chemical Physics, 2018, 148(24).
>
> [12] Desaphy J, Bret G, Rognan D, et al. sc-PDB: a 3D-database of ligandable binding sites—10 years on[J]. Nucleic acids research, 2015, 43(D1): D399-D404.

---

### Official Review · Reviewer_9vNs · 2023-10-31

**Soundness:** 3 good
**Presentation:** 3 good
**Contribution:** 3 good
**Rating:** 6
**Confidence:** 4

**Summary:**

In this paper, the authors proposed a new method, named EquiPocket, for ligand binding site prediction. Their model contains a local geometric modeling module, a global structure modeling module and a surface passing module to gather the surface geometric and multi-level structure features in a protein. Based on the experimental results, their method showed superiority, compared with other existing methods on several real data sets.

**Strengths:**

The idea of using an E(3)-equivariant GNN for binding site prediction is of interest especially in the bioinformatic domain.
Writing and presentation skill is well.
The proposed method is relative better than previous methods, which is not lack of significance.

**Weaknesses:**

[1] The authors did not compare their method with latest state-of-the-art methods, such as (1).
(1) Tubiana J, Schneidman-Duhovny D, Wolfson H J. ScanNet: an interpretable geometric deep learning model for structure-based protein binding site prediction[J]. Nature Methods, 2022, 19(6): 730-739.
[2] Some details need to be clarified. For example, proteins have multiple binding sites. How did the authors select binding sites.
[3] There is a significant correlation between binding sites and the properties of small molecules, not just spatial relationships.

**Questions:**

[1] The authors did not compare their method with latest state-of-the-art methods, such as (1).
(1) Tubiana J, Schneidman-Duhovny D, Wolfson H J. ScanNet: an interpretable geometric deep learning model for structure-based protein binding site prediction[J]. Nature Methods, 2022, 19(6): 730-739.
[2] Some details need to be clarified. For example, proteins have multiple binding sites. How did the authors select binding sites.
[3] There is a significant correlation between binding sites and the properties of small molecules, not just spatial relationships.
[4] The role of Dense Attention in reducing the negative impact caused by the protein size shift is limited.

---

> ### Author Response · Authors · 2023-11-19
> **Part 1/2 of Response to Reviewer  9vNs**
>
> We greatly appreciate your review, and in the following responses, we will address and provide explanations and additional information for the weaknesses and questions you have raised.
> >__Q1: The authors did not compare their method with latest state-of-the-art methods, such as (1). (1) Tubiana J, Schneidman-Duhovny D, Wolfson H J. ScanNet: an interpretable geometric deep learning model for structure-based protein binding site prediction[J]. Nature Methods, 2022, 19(6): 730-739.__
>
> For the ScanNet[9] method, we indeed explored this method during our research's early stages, as detailed in section A.3.3 of our paper. We attempted to compare EquiPocket with ScanNet but eventually opted against it due to the specific reasons:
>
> **Task Discrepancy.** EquiPocket is focused on predicting ligand binding sites, whereas ScanNet is designed for predicting binding sites in protein and protein, protein and antibody, and protein and disordered protein interactions, making it unsuitable for our specific task.
>
> **Incompatibility in Application.** ScanNet's pretrained model, dataset, and web service are tailored for its specific domains (details in provided [figures](https://z1.ax1x.com/2023/11/14/piYeQfS.png) and their [webserver](http://bioinfo3d.cs.tau.ac.il/ScanNet/index_real.html)). When applied to ligand binding site prediction, ScanNet yielded inconsistent results with the test datasets.
>
>
> >__Q2: Some details need to be clarified. For example, proteins have multiple binding sites. How did the authors select binding sites.__
>
> Thank you for your valuable guidance. The below table categorizes samples in test datasets based on the number of ligands binding to a protein. The majority of samples in the test dataset consist of a single binding site, with only a subset of samples containing multiple binding sites.
>
> | The number of ligands binding to a protein | COACH420 | Holo4k | PDBbind |
> |:----------------------------------------:|:--------:|:------:|:-------:|
> |                     1                    |    235   |  2442  |   5025  |
> |                     2                    |    36    |   635  |    0    |
> |                     3                    |     7    |   67   |    0    |
> |                     4                    |     4    |   22   |    0    |
> |                    >=5                   |     2    |   38   |    0    |
>
>
> Following related methods [4, 5, 6, 7, 8], the specific processing steps of our method are outlined below, and further details can be found in Section 4.1 and Appendix A.1 in our paper:
>
> **Training Process**:
>
> a. Following DeepSurf [7], the protein atoms within 4 Ångströms of any ligand atom are set as positive, otherwise negative.
>
> b. EquiPocket predicts the druggability probability for each atom, using the Dice loss function for model optimization.
>
> **Validation and Testing**:
>
> a. In accordance with related methods [4, 5, 6, 7, 8], we define the center of a binding site as the mean position of the ligand atoms. This definition allows us to handle the proteins with multiple binding sites (N >= 1).
>
> b. We focus on predicting druggability probabilities for each atom rather than identifying specific binding site atoms.
>
> c. Our method uses predicted atom-level probabilities in a clustering process [10] to autonomously identify ligand binding site centers and druggability, which are crucial for metrics and downstream docking tasks.
>
> **Performance Evaluation**:
>
>  a. We evaluate based on the top-$n$ predicted binding site centers, where $n$ does not exceed the actual number of binding sites. If no predicted binding sites center is found, it is considered a failure.
>
>  b. The predicted ligand binding center with the DCC/DCA falls below a predetermined threshold, typically set to 4, is classified as successful.
>
> Through the aforementioned process, we are able to effectively handle proteins with more than one binding site during training, validation, and testing.

---

> ### Author Response · Authors · 2023-11-19
> **Part 2/2 of Response to Reviewer  9vNs**
>
> >__Q3: There is a significant correlation between binding sites and the properties of small molecules, not just spatial relationships.__
>
> We fully understand your concerns. Indeed, a single binding site might bind to multiple ligands, with the binding interface varying based on each ligand's structure and properties. However, these variations minimally impact our main task. Ligand binding sites on a protein are generally seen as fixed regions, defined as a fixed-size box around a designated center [1, 2, 3, 4, 5, 6, 7, 8, 9]. This remains constant despite slight variations in how different ligands interact with the same binding site. More importantly, in our dataset as showed in table of Q2, most proteins have only one binding site, and usually, each binding site interacts with a single ligand. We have included the abve explanations in the revised paper to address your concern.
>
> >__Q4:  The role of Dense Attention in reducing the negative impact caused by the protein size shift is limited.__
>
> | Atom Num  | Protein Num | Ratio | Cumsum Ratio | DCC of EquiPocket(w/o attention) | DCC of EquiPocket |
> |-----------|-------------|-------|--------------|---------------------------|------------|
> | 0-1000    | 296         | 0.04  | 0.04         | 0.328                     | 0.428      |
> | 1000-2000 | 2193        | 0.27  | 0.3          | 0.547                     | 0.621      |
> | 2000-3000 | 2534        | 0.31  | 0.61         | 0.551                     | 0.590       |
> | 3000-4000 | 1143        | 0.14  | 0.75         | 0.343                     | 0.388      |
> | 4000-5000 | 619         | 0.08  | 0.82         | 0.261                     | 0.255      |
> | >=5000    | 1440        | 0.18  | 1            | 0.161                     | 0.153      |
>
> Thanks. We are sorry that the benefit of the Dese Attention module is not clearly demonstrated. We provide more evidence here. Our paper's Figure 6C compares EquiPocket with/without the Dense Attention module. We removed the Attention Module in EquiPocket (w/o attention) for this comparison. The above table presents results across different protein sizes:
>
> It is evident that Dense Attention notably improves prediction for proteins with less than 4000 nodes. However, for larger proteins, exceeding 4000 nodes, there's no significant performance difference.
> These findings highlight that Dense Attention boosts predictive accuracy for smaller proteins while maintaining performance for larger ones.
>
> **Reference**
>
> [1] Zhang Y, Cai H, Shi C, et al. E3bind: An end-to-end equivariant network for protein-ligand docking[J]. arXiv preprint arXiv:2210.06069, 2022.
>
> [2] Lu W, Wu Q, Zhang J, et al. Tankbind: Trigonometry-aware neural networks for drug-protein binding structure prediction[J]. Advances in neural information processing systems, 2022, 35: 7236-7249.
>
> [3] Liao Z, You R, Huang X, et al. DeepDock: enhancing ligand-protein interaction prediction by a combination of ligand and structure information[C]//2019 IEEE International Conference on Bioinformatics and Biomedicine (BIBM). IEEE, 2019: 311-317.
>
> [4] Krivák R, Hoksza D. P2Rank: machine learning based tool for rapid and accurate prediction of ligand binding sites from protein structure[J]. Journal of cheminformatics, 2018, 10: 1-12.
>
> [5] Le Guilloux V, Schmidtke P, Tuffery P. Fpocket: an open source platform for ligand pocket detection[J]. BMC bioinformatics, 2009, 10(1): 1-11.
>
> [6] Stepniewska-Dziubinska M M, Zielenkiewicz P, Siedlecki P. Improving detection of protein-ligand binding sites with 3D segmentation[J]. Scientific reports, 2020, 10(1): 5035.
>
> [7] Mylonas S K, Axenopoulos A, Daras P. DeepSurf: a surface-based deep learning approach for the prediction of ligand binding sites on proteins[J]. Bioinformatics, 2021, 37(12): 1681-1690
>
> [8] Jiménez J, Doerr S, Martínez-Rosell G, et al. DeepSite: protein-binding site predictor using 3D-convolutional neural networks[J]. Bioinformatics, 2017, 33(19): 3036-3042.
>
> [9] Tubiana J, Schneidman-Duhovny D, Wolfson H J. Scannet: A web server for structure-based prediction of protein binding sites with geometric deep learning[J]. Journal of Molecular Biology, 2022, 434(19): 167758.
>
> [10] Comaniciu D, Meer P. Mean shift: A robust approach toward feature space analysis[J]. IEEE Transactions on pattern analysis and machine intelligence, 2002, 24(5): 603-619.

---

> ### Author Response · Authors · 2023-11-22
> **Rebuttal Deadline Reminder**
>
> Dear reviewer 9vNs:
>
> Thank you very much for your review.
>
> I would like to kindly remind you that the rebuttal period is coming to an end. Could you please inform us if our responses have resolved your concerns, or if there are any other questions you need us to address?

---

### Official Review · Reviewer_HnAC · 2023-11-01

**Soundness:** 3 good
**Presentation:** 3 good
**Contribution:** 2 fair
**Rating:** 5
**Confidence:** 3

**Summary:**

This paper proposes a E(3)-equivariant graph neural network for ligand binding site prediction, where only the protein is given and the goal is to classify which protein atoms belong to the binding site. The model consists of a local structure modeling module, a global structure modeling module and a surface message passing module. The experiments show that the proposed method outperforms geometric-based method, CNN-based methods, 2D/3D graph-based methods with a clear margin.

**Strengths:**

- The proposed method shows clear superior empirical performance compared to existing methods.
- The paper is well-written and easy to follow
- The code is provided

**Weaknesses:**

The significance of the studied problem is limited, given lots of ligand docking prediction models and structure-based drug design models are proposed. It would be better if the authors could show how much improvement the proposed method can bring to the downstream tasks.

**Questions:**

- Why is relative direction prediction needed? The ablation study about this loss is needed.
- I'm a bit confused with the setting of this task. For the same protein, there may exist multiple ligands that can bind with it. Then, how are the ground-truth labels computed? If they are viewed as individual datapoints, one input protein may have multiple different sets of labels. Will it influence the training and prediction phase?

---

> ### Author Response · Authors · 2023-11-19
> **Part 1/2 of Response to Reviewer HnAC**
>
> Thank you so much for your valuable and insightful comments.
> We address your concern below and will contain the additional explanations in the revised paper.
> >__W1: The significance of the studied problem is limited, given lots of ligand docking prediction models and structure-based drug design models are proposed. It would be better if the authors could show how much improvement the proposed method can bring to the downstream tasks.__
>
> Thank you for the comment. As already demonstrated in previous studies [4, 5, 6, 7, 8], ligand binding site (or pocket) prediction comes as a fundamental step for many downstream tasks. For example, the models such as E3Bind [1], TankBind [2], and DeepDock [3] mainly focus on the tasks like protein-ligand binding pose and affinity prediction. They often use tools like P2rank [4] to identify target pocket for later fine-grained
> prediction of ligand docking pose. Our EquiPocket shares the same function as P2rank and can be applied in broad docking methods. It will be interesting to see the actual performance improvement in downstream tasks, which, however, requires end2end experimental evaluations and could be better left for future exploration.
>
>
> >__Q1: Why is relative direction prediction needed? The ablation study about this loss is needed.__
>
> The core reasons are as follows: Firstly, for ligand binding sites prediction task[4, 5, 6, 7, 8], the ligand cannot be used as input data and can only be employed to define the target. Using it as input would risk data leakage. Secondly, the relative direction between a protein atom and its nearest ligand atom effectively captures the local geometric information of the binding sites on a protein. Different from the previous works [6, 7, 8], our method can output E(3)-equivariant coordinates (detailed in Section 3.3 of our paper). To better capture the geometric details of the protein surface, we introduced more detailed relative direction as a supplementary target.
>
> The corresponding ablation results have been shown in Figure 6 (c) of our original paper, and detailed results are presented below.
> EquiPocket (w/o Direction Loss) represents our EquiPocket method removing the relative direction prediction module.
> It can be observed that when the relative direction prediction module is removed, our method's performance drops for proteins of different sizes. This is especially notable for proteins with fewer than 3000 atoms, which account for 60% of the samples. If the relative direction prediction is removed, the performance drops by approximately 10%.
> These results demonstrate the effectiveness of our designed relative direction target.
>
> | Atom Num  | Protein Num | Ratio | Cumsum Ratio | DCC of EquiPocket(w/o Direction loss) | DCC of EquiPocket |
> |-----------|-------------|-------|--------------|--------------------------------|------------|
> | 0-1000    | 296         | 0.04  | 0.04         | 0.319                          | 0.428      |
> | 1000-2000 | 2193        | 0.27  | 0.3          | 0.587                          | 0.621      |
> | 2000-3000 | 2534        | 0.31  | 0.61         | 0.551                          | 0.590       |
> | 3000-4000 | 1143        | 0.14  | 0.75         | 0.371                          | 0.388      |
> | 4000-5000 | 619         | 0.08  | 0.82         | 0.258                          | 0.255      |
> | >=5000    | 1440        | 0.18  | 1            | 0.144                          | 0.153      |

---

> ### Author Response · Authors · 2023-11-19
> **Part 2/2 of Response to Reviewer HnAC**
>
> >__Q2:I'm a bit confused with the setting of this task. For the same protein, there may exist multiple ligands that can bind with it. Then, how are the ground-truth labels computed? If they are viewed as individual datapoints, one input protein may have multiple different sets of labels. Will it influence the training and prediction phase?__
>
> Your perspective is indeed accurate. The below table categorizes samples in test datasets based on the number of ligands binding to a protein. The majority of samples in the test dataset consist of a single binding site, with a subset of samples contains multiple binding sites.
>
> | The number of ligands binding to a protein | COACH420 | HOLO4k | PDBbind |
> |:----------------------------------------:|:--------:|:------:|:-------:|
> |                     1                    |    235   |  2442  |   5025  |
> |                     2                    |    36    |   635  |    0    |
> |                     3                    |     7    |   67   |    0    |
> |                     4                    |     4    |   22   |    0    |
> |                    >=5                   |     2    |   38   |    0    |
>
>
> Following related methods [4, 5, 6, 7, 8], the specific processing steps of our method are outlined below, and the further details can be find in Section 4.1 and Appendix A.1 of our paper:
>
>
> **Training Process**:
>
> a. Following DeepSurf [7], the protein atoms within 4 Ångströms of any ligand atom are set as positive and negative otherwise.
>
> b. EquiPocket predicts the druggability probability for each atom, using the Dice loss function for model optimization.
>
> **Validation and Testing**:
>
> a. In accordance with related methods[4, 5, 6, 7, 8], we define the center of a binding site as the mean position of the ligand atoms. This definition allows us to handle the proteins with multiple binding sites (N >= 1).
>
> b. We focus on predicting druggability probabilities for each atom rather than identifying specific binding site atoms.
>
> c. Our method uses predicted atom-level probabilities in a clustering process [10] to autonomously identify ligand binding site centers and druggability, which are crucial for metrics and downstream docking tasks.
>
> **Performance Evaluation**:
>
>  a. We evaluate based on the top-n predicted binding site centers, where n does not exceed the actual number of binding sites. If no predicted binding sites center is found, it is considered a failure.
>
>  b. The predicted ligand binding center with the DCC/DCA falls below a predetermined threshold, typically set to 4, is classified as successful.
>
> Through the aforementioned process, we are able to effectively handle proteins with more than one binding site during training, validation, and testing.
>
>
>
> **Reference**
>
> [1] Zhang Y, Cai H, Shi C, et al. E3bind: An end-to-end equivariant network for protein-ligand docking[J]. arXiv preprint arXiv:2210.06069, 2022.
>
> [2] Lu W, Wu Q, Zhang J, et al. Tankbind: Trigonometry-aware neural networks for drug-protein binding structure prediction[J]. Advances in neural information processing systems, 2022, 35: 7236-7249.
>
> [3] Liao Z, You R, Huang X, et al. DeepDock: enhancing ligand-protein interaction prediction by a combination of ligand and structure information[C]//2019 IEEE International  Conference on Bioinformatics and Biomedicine (BIBM). IEEE, 2019: 311-317.
>
> [4] Krivák R, Hoksza D. P2Rank: machine learning based tool for rapid and accurate prediction of ligand binding sites from protein structure[J]. Journal of cheminformatics, 2018, 10: 1-12.
>
> [5] Le Guilloux V, Schmidtke P, Tuffery P. Fpocket: an open source platform for ligand pocket detection[J]. BMC bioinformatics, 2009, 10(1): 1-11.
>
> [6] Stepniewska-Dziubinska M M, Zielenkiewicz P, Siedlecki P. Improving detection of protein-ligand binding sites with 3D segmentation[J]. Scientific reports, 2020, 10(1): 5035.
>
> [7] Mylonas S K, Axenopoulos A, Daras P. DeepSurf: a surface-based deep learning approach for the prediction of ligand binding sites on proteins[J]. Bioinformatics, 2021, 37(12): 1681-1690
>
> [8] Jiménez J, Doerr S, Martínez-Rosell G, et al. DeepSite: protein-binding site predictor using 3D-convolutional neural networks[J]. Bioinformatics, 2017, 33(19): 3036-3042.
>
> [9] Tubiana J, Schneidman-Duhovny D, Wolfson H J. Scannet: A web server for structure-based prediction of protein binding sites with geometric deep learning[J]. Journal of Molecular Biology, 2022, 434(19): 167758.
>
> [10] Comaniciu D, Meer P. Mean shift: A robust approach toward feature space analysis[J]. IEEE Transactions on pattern analysis and machine intelligence, 2002, 24(5): 603-619.

---

> ### Author Response · Authors · 2023-11-20
> **Rebuttal Deadline Reminder**
>
> Dear reviewer HnAC:
>
> I would like to kindly remind you that the rebuttal period is coming to an end. Could you please inform us if our responses have resolved your concerns, or if there are any other questions you need us to address?

---

> > ### Comment · Reviewer_HnAC · 2023-11-21
> >
> > Thank you for the authors' response. My concerns about relative direction loss and multiple binding sites training / inference are addressed, but other concerns still remain. Specifically, the authors highlight that their method shares the same function as P2rank, but didn't compare with it and didn't conduct experiments on downsteaming tasks (such as docking) to show how much improvement the proposed model can bring, which weakens this work's significance. Thus, I'm leaning toward the borderline reject.

---

> > > ### Author Response · Authors · 2023-11-22
> > > **Response to Reviewer HnAC**
> > >
> > > Dear reviewer HnAC:
> > >
> > > Thank you very much for your reply. We have studied P2Rank[4] in our paper (Section A3.1). However, we didn't include P2Rank as a baseline mainly due to the differences in training and validation data between it and deep learning methods including Kalasanty [1], DeepSurf [2], deepsite [3], and our EquiPocket. Specifically, P2Rank uses data from CHEN11 and JOINED datasets, while deep learning methods commonly use scPDB[5]. P2Rank's paper mentions (details in [here](https://z1.ax1x.com/2023/11/15/piYcPNn.png)) that CHEN11 is more diverse than scPDB , affecting model performance.
> > > **We have enriched the comparison between our method and P2rank in Appendix A.3.1 of our paper.**
> > >
> > > Logically speaking, most current methods[1, 2, 3] for ligand binding site prediction require voxelization, facing challenges like sensitivity to rotations and insensitivity to protein size shifts. Our EquiPocket method effectively overcomes these issues, enhancing prediction accuracy and positively impacting downstream tasks:
> > >
> > > a. Improved Docking Efficiency. The high precision of binding site prediction of our method can reduce the number of candidate binding sites needed for protein-ligand docking,thereby lowering computational costs.
> > >
> > > b. Atom-Level Drugability Predictions. Our method provides detailed predictions on the probability of atoms being part of a binding site, enhancing the prediction of protein structure, binding pose, affinity, and more.
> > >
> > > c. Complementing AlphaFold2[11]: With AlphaFold2 predicting structures for millions of proteins, many still lack detailed binding site information. EquiPocket can help identify and understand these structures and their binding sites, aiding drug research efforts.
> > >
> > >
> > >
> > > **Reference**
> > >
> > > [1] Stepniewska-Dziubinska M M, Zielenkiewicz P, Siedlecki P. Improving detection of protein-ligand binding sites with 3D segmentation[J]. Scientific reports, 2020, 10(1): 5035.
> > >
> > > [2] Mylonas S K, Axenopoulos A, Daras P. DeepSurf: a surface-based deep learning approach for the prediction of ligand binding sites on proteins[J]. Bioinformatics, 2021, 37(12): 1681-1690
> > >
> > > [3] Jiménez J, Doerr S, Martínez-Rosell G, et al. DeepSite: protein-binding site predictor using 3D-convolutional neural networks[J]. Bioinformatics, 2017, 33(19): 3036-3042.
> > >
> > > [4] Krivák R, Hoksza D. P2Rank: machine learning based tool for rapid and accurate prediction of ligand binding sites from protein structure[J]. Journal of cheminformatics, 2018, 10: 1-12.
> > >
> > > [5]Desaphy J, Bret G, Rognan D, et al. sc-PDB: a 3D-database of ligandable binding sites—10 years on[J]. Nucleic acids research, 2015, 43(D1): D399-D404.

---

### Official Review · Reviewer_qDNY · 2023-11-04

**Soundness:** 3 good
**Presentation:** 4 excellent
**Contribution:** 3 good
**Rating:** 6
**Confidence:** 4

**Summary:**

This work proposes a novel method for protein binding site predictions.
To address the limitations of previous approaches (e.g., relying on voxelization, insensitive to se3 transformation, unaware of protein size, insufficient characterization of protein surface), the authors propose to model both the local geometrical features of surface atoms and global protein features based on chemical graphs and spatial graphs.

To account for variable protein sizes, they also devise a novel dense attention layer to aggregate representations of different encoding layers based on their importance.

They conduct extensive experiments including ablation studies to validate the effectiveness of the method.

**Strengths:**

1. The paper summarizes several common limitations of previous works. It seems the proposed method fairly addresses these issues according to numerical results and the story presented in the paper.

2. Modeling surface features of target proteins is of crucial importance to the success of the binding site prediction. This paper gives insights into modeling surface features. In specific, they propose a way to construct surface graphs and corresponding node features given spatial information of proteins. They also adapt the EGNN into surface-EGNN, which seems to be effective at modeling both the chemical and geometrical features of target proteins.

3. To account for the variable protein sizes, the authors propose a dense attention output layer to aggregate features from different layers according to their importance. They show via ablation study that the module is crucial to the performance.

**Weaknesses:**

1. My main concern is that the proposed method is unaware of the structure of ligand molecules (or proteins). In a real-world setting, the binding site of a target protein might be dependent on the ligands they bind to.

2. P2rank is a widely used package in literature (e.g., TANKBIND[1], E3BIND[2]) for locating the ligand-binding pockets based on protein structures. It is quite accurate and efficient. The authors should include it as a baseline method.

**Questions:**

1. What is the rationale behind Eq.3? Why do you swap the order or MLP and Pooling?
2. Can the proposed method be adapted to protein-protein binding site prediction?

---

> ### Author Response · Authors · 2023-11-19
> **Part 1/2 of Response to Reviewer qDNY**
>
> Thank you very much for your supportive and constructive comments.  We provide point-by-point responses to your concerns as follows.
>
>  >__W1: My main concern is that the proposed method is unaware of the structure of ligand molecules (or proteins). In a real-world setting, the binding site of a target protein might be dependent on the ligands they bind to. .__
>
> We fully understand your concerns, and there may be some misunderstanding in this context.
>
> The task of our EquiPocket is ligand binding site prediction. **The key output of our method is identifying the center of candidate binding sites on the protein, rather than detailing the exact protein-ligand interface**.  The area around the center is seen as candidate binding sites for further docking tasks, and it is unnecessarily related to specific ligands. For example, the docking methods such as E3Bind [1] and TankBind [2] first use existing methods to predict the binding site of the protein, and then conduct docking process by considering the interaction between the ligand and the protein  around the binding site.
>
> The below table categorizes samples in test datasets based on the number of ligands binding to a protein.
> Indeed, most proteins have only one ligand binding site, though some can bind with multiple ligands. The binding interfaces depend on the structure and properties of each ligand. However, these variations minimally impact the task. Ligand binding sites on a protein are generally seen as fixed regions, defined as a fixed-size box around a designated center[1, 2, 3, 4, 5, 6, 7, 8, 9]. This remains constant despite slight variations when different ligands interact with the same binding site.
>
> | The number of ligands binding to a protein | COACH420 | HOLO4k | PDBbind |
> |:----------------------------------------:|:--------:|:------:|:-------:|
> |                     1                    |    235   |  2442  |   5025  |
> |                     2                    |    36    |   635  |    0    |
> |                     3                    |     7    |   67   |    0    |
> |                     4                    |     4    |   22   |    0    |
> |                    >=5                   |     2    |   38   |    0    |
>
> > __W2: P2rank is a widely used package in literature (e.g., TANKBIND[1], E3BIND[2]) for locating the ligand-binding pockets based on protein structures. It is quite accurate and efficient. The authors should include it as a baseline method.__
>
> Thank you for your suggestions. We have studied P2Rank [4] in our paper (Section A3.1). However, we didn't include P2Rank as a baseline mainly due to **the differences in training and validation data between it and deep learning methods including Kalasanty [6], DeepSurf [7], deepsite [8], and our EquiPocket.** Specifically, P2Rank uses data from CHEN11 and JOINED datasets, while deep learning methods commonly use scPDB[10].P2Rank's paper mentions (details in [here](https://z1.ax1x.com/2023/11/15/piYcPNn.png)) that CHEN11 is more diverse than scPDB , affecting model performance. The comparison results are as following:
>
> | DCA                | COACH420 | HOLO4K |
> |--------------------|----------|--------|
> | P2Rank[protrusion] | 0.642    | 0.593  |
> | P2rank             | 0.683    | 0.706  |
> | DeepSite           | 0.564    | 0.456  |
> | Kalasanty          | 0.636    | 0.515  |
> | deepsurf           | 0.658    | 0.635  |
> | EquiPocket         | 0.656    | 0.662  |
>
> The results in above table show that our method essentially matches or surpasses most deep learning methods, even outperforming P2Rank [protrusion], which uses only geometric information, and slightly trailing behind P2Rank, which benefits from a more diverse dataset.
> **These results has been added to the revised paper.**

---

> ### Author Response · Authors · 2023-11-19
> **Part 2/2 of Response to Reviewer qDNY**
>
> >__Q1: What is the rationale behind Eq.3? Why do you swap the order or MLP and Pooling?__
>
> Eq.3: $g_{i}=[Pooling( \\{ MLP(g(s_{j})) \\} \_{s_{j} \in S_{i}} ), MLP(Pooling(\\{g(s_{j})\\}\_{s_{j} \in S_{i}}))]$
>
> In Eq.3, $g_i$ is the geometric embedding for a protein atom, learned from surrounding surface probes, and $s_i$ denotes the local geometric properties of these probes with properties such as distances and angle to protein atoms, the surface center, neighboring probes, and so on.
> Initially, we apply an MLP to these features, followed by pooling. This process transforms the geometric properties before aggregating them into the protein node's geometric embedding.
> However, since each property of  $s_i$ itself carries meaningful information. We are concerned that applying MLP first and then pooling might weaken the transmission of this information. Therefore, we take the second part of the equation.
>
> |                   |            | Coach420 |        | HOLO4k |        | PDBbind |        |
> |:-----------------:|:----------:|:--------:|:------:|:------:|:------:|:-------:|:------:|
> |       Model       | Fail Ratio |    DCC   |   DCA  |   DCC  |   DCA  |   DCC   |   DCA  |
> | EquiPocket-former |    0.16    |  0.389   | 0.606  | 0.330  | 0.637  |  0.507  | 0.660  |
> | EquiPocket-latter |    0.16    |  0.407   | 0.617  | 0.319  | 0.644  |  0.529  | 0.676  |
> |     EquiPocket    |    0.05    |  0.423   | 0.656  | 0.337  | 0.662  |  0.545  | 0.721  |
>
> To highlight the effectiveness of the features in Eq.3, we carried out extra experiments with "EquiPocket-former" focusing on the equation's initial part and "EquiPocket-latter" on its latter part. The findings show:
> Using either feature alone diminishes the predictive performance compared to the full EquiPocket model. Specifically, "EquiPocket-former" alone sees about a 10% drop, while "EquiPocket-latter" alone results in around a 5% reduction.
> This outcome underscores the necessity of both features, with the latter part having a more substantial impact on our model’s performance.
>
> >__Q2: Can the proposed method be adapted to protein-protein binding site prediction?__
>
> Nice suggestion. However, in our current version, we focus only on ligand-protein binding site prediction for these reasons:
>
> a. Our method is specifically trained on a dataset for ligand binding sites [10]. Therefore, directly applying it for protein-protein binding site prediction is not feasible due to the specialized nature of the training.
>
> b, While predicting protein-ligand and protein-protein binding sites may have some similarities, the differences in the types of interacting partners (ligands vs. proteins), their sizes, properties, and their effects on the target protein are significant. These differences likely require distinct model and feature design.
>
> We respectfully understand the reviewer's suggestion and are willing to extend the application scope of our method in future exploration.
>
>
> **Reference**
>
> [1] Zhang Y, Cai H, Shi C, et al. E3bind: An end-to-end equivariant network for protein-ligand docking[J]. arXiv preprint arXiv:2210.06069, 2022.
>
> [2] Lu W, Wu Q, Zhang J, et al. Tankbind: Trigonometry-aware neural networks for drug-protein binding structure prediction[J]. Advances in neural information processing systems, 2022, 35: 7236-7249.
>
> [3] Liao Z, You R, Huang X, et al. DeepDock: enhancing ligand-protein interaction prediction by a combination of ligand and structure information[C]//2019 IEEE International Conference on Bioinformatics and Biomedicine (BIBM). IEEE, 2019: 311-317.
>
> [4] Krivák R, Hoksza D. P2Rank: machine learning based tool for rapid and accurate prediction of ligand binding sites from protein structure[J]. Journal of cheminformatics, 2018, 10: 1-12.
>
> [5] Le Guilloux V, Schmidtke P, Tuffery P. Fpocket: an open source platform for ligand pocket detection[J]. BMC bioinformatics, 2009, 10(1): 1-11.
>
> [6] Stepniewska-Dziubinska M M, Zielenkiewicz P, Siedlecki P. Improving detection of protein-ligand binding sites with 3D segmentation[J]. Scientific reports, 2020, 10(1): 5035.
>
> [7] Mylonas S K, Axenopoulos A, Daras P. DeepSurf: a surface-based deep learning approach for the prediction of ligand binding sites on proteins[J]. Bioinformatics, 2021, 37(12): 1681-1690
>
> [8] Jiménez J, Doerr S, Martínez-Rosell G, et al. DeepSite: protein-binding site predictor using 3D-convolutional neural networks[J]. Bioinformatics, 2017, 33(19): 3036-3042.
>
> [9] Tubiana J, Schneidman-Duhovny D, Wolfson H J. Scannet: A web server for structure-based prediction of protein binding sites with geometric deep learning[J]. Journal of Molecular Biology, 2022, 434(19): 167758.
>
> [10] Desaphy J, Bret G, Rognan D, et al. sc-PDB: a 3D-database of ligandable binding sites—10 years on[J]. Nucleic acids research, 2015, 43(D1): D399-D404.

---

> ### Author Response · Authors · 2023-11-22
> **Rebuttal Deadline Reminder**
>
> Dear reviewer qDNY:
>
> Thank you very much for your review.
>
> I would like to kindly remind you that the rebuttal period is coming to an end. Could you please inform us if our responses have resolved your concerns, or if there are any other questions you need us to address?

---

### Author Response · Authors · 2023-11-19
**General Response**

We express our gratitude to all the reviewers for their constructive comments. Our paper has been thoroughly revised, highlighting modifications in red, to address the raised concerns. The primary revisions include:
1. In response to reviewer fzz6 , we expanded the discussion of ligand binding site predictions versus docking tasks in Section 1 and added Appendix A.6 and A.7 with information on method inference speeds and ablation studies for two feature extractors.
2. In response to reviewer 9vNs, we expanded the discussion on ScanNet and detailed experimental results of Dense Attention in Appendix A.3.3 and A.8.
3. In response to reviewer qDNY, we added comparison results for Eq.3 in Appendix A.10.
4. In response to reviewer HnAC, we added detailed experimental results of Relative Direction in Appendix A.9.
5. In response to reviewers HnAC and 9vNs, we enriched Appendix A.2 with a comprehensive summary of our test dataset and detailed experimental procedures.
6. In response to reviewers qDNY, HnAC and fzz6, we enriched the comparison and detailed results between our method and P2rank in Appendix A.3.1.


We hope these revisions resolve reviewer concerns and improve our paper's quality.

---

### Meta-Review · Area_Chair_g9y6 · 2023-12-09

**Metareview:**

This paper studies ligand binding site prediction with geometric deep learning. This is an emerging area with growing recent work. The main concern after rebuttals is comparison with recent methods to demonstrate the advantage of the proposed method. Thus a reject is recommended so that the authors can revise the paper accordingly.

**Justification For Why Not Higher Score:**

The main concern after rebuttals is comparison with recent methods to demonstrate the advantage of the proposed method.

**Justification For Why Not Lower Score:**

NA

---

### Decision · Program_Chairs · 2024-01-16

Reject